



**Overview towards improved understanding of the mechanisms leading to heavy**
**precipitation in the Western Mediterranean: lessons learned from HyMeX**
[1,2]Samira Khodayar, [3]Silvio Davolio, [4]Paolo Di Girolamo, [5]Cindy Lebeaupin Brossier,
[6]Emmanouil Flaounas, [5]Nadia Fourrie, [7,8] Keun-Ok Lee, [5]Didier Ricard, [5]Benoit Vie,
[5]Francois Bouttier, [2]Alberto Caldas-Alvarez, [5]Veronique Ducrocq
[1]Mediterranean Centre for Environmental Studies (CEAM), Valencia, Spain
[2]Institute of Meteorology and Climate Research (IMK-TRO), Karlsruhe Institute of Technology (KIT),
Karlsruhe, Germany
[3]National Research Council of Italy, Institute of Atmospheric Sciences and Climate, (CNR-ISAC), Bologna,
Italy
[4]Scuola di Ingegneria, Università degli Studi della Basilicata (SI-UNIBAS), Potenza, Italy
[5]CNRM, Université de Toulouse, Météo-France, CNRS, Toulouse, France
[6]Institute of Oceanography, Hellenic Centre for Marine Research (HCMR), Athens, Greece
[7]Laboratoire d'Aérologie, Université de Toulouse, CNRS, UPS, Toulouse, France
[8]Laboratoire de L'Atmosphère et des Cyclones, UMR 8105 (CNRS, Université de La Réunion, Météo-
France), Saint Denis, France
* Corresponding author. E-mail address: khodayar_sam@gva.es (S. Khodayar)
Mediterranean Centre for Environmental Studies (CEAM),
Technological Park, Charles R. Darwin Street, 14 46980 - Paterna - Valencia - Spain


**Abstract**
Heavy precipitation (HP) constitutes a major meteorological threat in the western
Mediterranean (WMed). Every year, recurrent events affect the area with fatal
consequences on infrastructure and personal losses. Despite this being a well-known
issue, widely investigated in the past, still open questions remain. Particularly, the
understanding of the underlying mechanisms and the modelling representation of the
events must be improved. One of the major goals of the Hydrological cYcle in the
Mediterranean eXperiment (HyMeX; 2010-2020) has been to advance knowledge on this
topic. In this article we present an overview of the most recent lessons learned from
HyMeX towards improved understanding of the mechanisms leading to HP in the WMed.
The unique network of instruments deployed, the use of finer model resolutions and of
coupled models, provided an unprecedented opportunity to validate numerical model
simulations, to develop improved parameterizations, designing high-resolution ensemble
modelling approaches and sophisticated assimilation techniques across scales.
All in all, HyMeX and particularly the science team heavy precipitation favoured the
evidencing of theoretical results, the enrichment of our knowledge on the genesis and
evolution of convection in a complex topography environment, and the improvement of
precipitation forecasts. Illustratively, the intervention of cyclones and warm conveyor
belts in the occurrence of heavy precipitation has been pointed out, the crucial role of
the spatio-temporal distribution of the atmospheric water vapor for the understanding
and accurate forecast of the timing and location of deep convection has been evidenced,
as well as the complex interaction among processes across scales. The importance of
soil and ocean conditions and the interactions among systems were highlighted and such
systems were specifically developed in the framework of HyMeX to improve the realism
of weather forecasts. Furthermore, the benefits of cross-disciplinary efforts within HyMeX
have been a key asset in bringing a step forward our knowledge about heavy
precipitation in the Mediterranean region.


## 1. Introduction and Motivation

A 10-Year Multidisciplinary Program on the Mediterranean Water Cycle, HyMeX, Hydrological Cycle in the Mediterranean Experiment (Drobinski et al., 2014), has come to an end (2010-2020). With the main goal to advance the scientific knowledge of the Mediterranean water cycle variability and to improve the process-based and regional climate models, different temporal scales are considered, from weather-scale to the seasonal and interannual scales. Special focus is put on the hydrometeorological extremes and consequent social and economic impacts, as well as the vulnerability and the adaptation capacity of the Mediterranean population under Climate Change.

The unique character of the Mediterranean basin and surrounding countries resulting from the geographical location, climatic conditions, and topography, makes the region prone to extreme phenomena; heavy precipitation and flash floods, as well as heat waves and drought (e.g., Mariotti, 2010). The region, defined as one of the two main "hot spots" of climate change (Giorgi, 2006; IPCC, 2013), is in a transition area, therefore, very sensitive to global climate change at short and long-time scales. An increase in interannual rainfall variability, strong warming and drying in addition to a significant population growth are projected for the coming future. Despite the overall Mediterranean climate drying, under climate change, intensity of heavy precipitation events (HPEs) is expected to increase (Planton et al., 2016; Jacob et al., 2013; Drobinski et al., 2016; Colmet-Daage et al., 2017; Tramblay and Somot, 2018; Giorgi et al., 2019). In this context, threats posed from the expected increase in frequency and intensity of events conducive to floods and droughts (Gao et al., 2006; Orlowsky and Seneviratne, 2011) are seen with great concern. Countries surrounding the Mediterranean basin already suffer water problems in relation to water shortages and floods. Food security could also become an issue (Nelson et al., 2010).

HPEs and the associated flash floods are the most dangerous meteorological hazards affecting the Mediterranean countries in terms of mortality, and hundreds of millions of euros in damages are registered every year (Llasat et al., 2010, 2013; Doocy et al., 2013). The Mediterranean basin and particularly the surrounding mountainous coastal regions are often affected by these phenomena, regularly in the autumn period. The Mediterranean Sea, acting as a heat and moisture source, and the steep orography, triggering convection, are key aspects determining the occurrence of heavy precipitation in the region which is mainly of convective nature (Funatsu et al., 2008; Dayan et al., 2015). Rainfall accumulations greater than 100-150 mm may be expected in less than a day or even just a few hours resulting mostly from quasi-stationary mesoscale convective



systems (MCSs; Lee et al., 2018, 2017, 2016; Duffourg et al 2018; Buzzi et al., 2014).
Such rainfall accumulations are favoured by a slowly evolving synoptic situation,
characterized by an upper-level trough and consequent cyclogenesis that induces
advection of warm and moist air from the Mediterranean Sea (Duffourg and Ducrocq,
2011) to the coasts through marine low-level jets (Homar et al., 1999; Jansa et al., 2001;
Nuissier et al., 2011; Ricard et al., 2012; Khodayar et al., 2016b). Strong wind with high
sea surface temperature (SST) governs evaporation, which moistens and warms the
lowest levels of the atmosphere, thus increasing instability and finally often enhancing
the convection intensity (Xie et al., 2005, Lebeaupin et al., 2006; Stocchi and Davolio,
2017, Rainaud et al., 2017; Senatore et al., 2020a). Low-level convergence over the sea,
cold pools beneath the convective systems, or topographic lifting when encountering the
coastal mountains trigger deep convection, forcing the lift of the conditionally unstable
low-level flow. The synoptic-scale situations associated with these episodes are
generally well-known and well represented in numerical weather prediction (NWP) model
simulations. However, the accuracy of forecasts is still insufficient to adequately assess
timing, location and intensity of rainfall and flash flooding in certain situations, which is a
key step towards prevention and mitigation. This is mostly in relation to (a) model
limitations in terms of predictability of small-scale processes (e.g., convection,
turbulence) and feedbacks (e.g., soil, atmosphere, ocean) and their non-linear
interaction across scales, (b) lack of knowledge regarding underlying mechanisms, and
(c) absence of adequate observations to help us advance our understanding and
improving model capabilities.
This issue is one of the main objectives of the HyMeX international programme, and of
its associated first special observation period (SOP1; Ducrocq et al., 2014), from 5
September to 6 November 2012, dedicated to heavy precipitation and flash flooding.
Because of the large number of instruments deployed, the unprecedented high spatial-
temporal coverage achieved and the quality of the derived observations, the SOP1 has
offered a unique opportunity to improve understanding and advance documenting high-
impact weather events. This is in addition to the significant progress achieved in the last
decade through the development of convection-permitting models, whose benefit has
been sufficiently demonstrated (Richard et al., 2007; Fosser et al., 2014, Prein et al.,
2015; Clark et al, 2016, among others) and it is widely used nowadays from the scientific
community.
The major goal of this article is to expose an overview on some of the recent years' main
achievements towards better understanding of the mechanisms leading to heavy
precipitation in the WMed in the framework of the HyMeX international programme.
Advances regarding improved understanding of the mechanisms governing the initiation



and intensification of precipitating systems producing large amounts of rainfall are
thoroughly discussed in terms of in situ observations and high-resolution modelling
systems, as well as the synergetic use of both to help us bridging knowledge gaps. An
intensive observation period IOP16, which took place during the SOP1, is taken as a
paradigm to illustrate some of the main HyMeX results in the field of heavy precipitation.
This paper is structured as follows: in section 2 we describe the general conditions
leading to HP during the SOP1 period, the state-of-the-art of the observational networks
deployed in this time, as well as the modelling strategy developed. Additionally, the
IOP16, which has been used throughout the paper for illustrating some of the results is
presented. In section 3, the main advances regarding HP understanding and modelling
are presented, including the large-scale dynamics, advances in moist process
understanding, low-level dynamics, the impacts of the land and the sea surfaces and
microphysics. Section 4 is devoted to the examination of the improvements in the multi-
scale modelling of HP and in section 5 some conclusions and recommendations are
summarized.

## 2. Heavy precipitation during the HyMeX SOP1 period

The SOP1 campaign took place in 2012, from 5 September to 6 November, when the
probability of HPE occurrence in the north-western Mediterranean is the highest. About
30% of the days in this period experienced, indeed, rainfall accumulations over 100 mm
somewhere in the investigation domain. Sixteen Intensive Observation Periods (IOPs)
were launched during the campaign, most of them occurring in the period after mid-
October to the end of the SOP1 (Ducrocq et al., 2014). This agrees with the monthly
precipitation totals being close to the climatological values in September, but well above
in October (Khodayar et al., 2016b). Most IOPs did not affect a single site but
encompassed several regions of the north-western Mediterranean. The most affected
sites were the Cévennes-Vivarais (CV), including the Massif Central and the French
Southern Alps, as well as the Liguria-Tuscany (LT) region in Italy.

### 2.1. State-of-the-art observational capabilities and modelling activities

More than 200 research instruments were deployed over the WMed Sea and surrounding
countries, namely Spain, France, and Italy to ensure a close observation of the
precipitating systems and a fine-scale survey of the upstream meteorological conditions
over the Mediterranean. Ducrocq et al. (2014) provides a comprehensive description of
the observing systems deployed during the SOP1. Furthermore, this unique network of
instruments provided an unprecedented opportunity to validate more accurately NWP
model simulations, to develop novel data assimilation techniques and to improve model





parameterizations with the purpose of better predicting the evolution of the environment
across scales.

### 2.1.1 Ground-based, airborne, and seaborne observations

One unique aspect of HyMeX-SOP1 was represented by the availability of a large
ensemble of ground-based and airborne instruments, covering a major portion of the
WMed and its surrounding coastal regions in France, Italy, and Spain. The observational
domain of HyMeX-SOP1 was defined to include the area with the highest occurrence of
HPEs and being within the ranges of aircraft flight endurance. Within this large domain,
five measurement sites including advanced research instruments were established, i.e.
the Cévennes-Vivarais (CV) and the Corsica (CO) sites, the Central Italy (CI) and
Northeastern Italy (NEI) sites, and the Spanish Balearic Islands (BA) site in Menorca
(Ducrocq et al., 2014). Most sites were equipped with soil moisture sensors, turbulence
or energy balance stations, microwave radiometers, lidars, radars (cloud, precipitation
and/or wind) and radiosonde launching facilities, in addition to the operational
meteorological and hydrological ground networks covering the entire SOP1 domain.
Thus, an unprecedented dense network of rain gauges was available over France, Italy,
and Spain, with a density of about one hourly rain gauge per 180 km$^2$. This network
operated in combination with a radar network including a variety of S-band, C-band
Doppler (two of them being polarimetric) and X-band radars (one of them being
polarimetric). A similarly dense network of Global Positioning System (GPS) stations was
also established, with stations covering the north-western Mediterranean basin and
including measurements from 25 European, national, and regional GPS networks (Bock
et al., 2016).
Three aircrafts participated in the field campaign: the French ATR42, the French Falcon
20 (operated by SAFIRE (Service des Avions Français Instrumentés pour la Recherche
en Environnement)) and the German Do128 (Corsmeier et al., 2001). The ATR42
involvement was primarily aimed to characterize the origin and transport pattern of water
vapor and aerosol in pre-convective conditions and their link with heavy precipitating
systems. Its main payload was the airborne dial LEANDRE 2, capable of profiling water
vapor mixing ratio above or beneath the aircraft. The F20 aircraft primary mission was
the characterization of the microphysical and kinematic processes taking place within
convective precipitating systems, this objective being pursued based on the use of
advanced microphysical in situ probes and the 95-GHz Doppler cloud radar RASTA
(Radar Aéroporté et Sol de Télédétection des propriétés nuAgeuses, Protat et al., 2009).
Furthermore, the German Do 128 research aircraft was equipped with fast sensors to
measure turbulent fluxes, water vapor inlet, and stable water isotope measurements
(Sodemann et al., 2017) with the primary goal of monitoring upstream low-level





conditions before and during HPEs and investigating the orographic and thermal impact
of the island on the initiation and evolution of diurnal convective activity.
During HyMeX-SOP1 Boundary Layer Pressurized Balloons (BLPBs) were also
launched from Menorca, flying at a nearly constant height (Doerenbecher et al., 2016)
and providing Lagrangian trajectories of specific humidity, temperature, pressure, and
horizontal wind.
Two ground-based Raman lidars were involved, namely the system BASIL (Di Girolamo
et al., 2009), deployed in Candillargues (Southern France) and the system WALI (Water-
vapour Raman Lidar; Chazette et al., 2016), deployed in Ciutadella (Menorca, Balearic
Islands). Both systems provided long-term records of high-resolution and accurate
humidity measurements, both in daytime and night-time, throughout the duration of
HyMeX-SOP1.
At sea, several platforms were deployed to monitor the ocean upper-layer and the
exchanges with the atmosphere (Ducrocq et al., 2014, Lebeaupin Brossier et al., 2014,
Rainaud et al., 2015). Two Météo-France moored buoys, LION (4.7°E−42.1°N) and
AZUR (7.8°E−43.4°N), routinely provide the 2 m-temperature, humidity, 10 m-wind
speed, direction and gust intensity, mean sea level pressure and sea surface parameters
(SST, wave height and period). They were equipped with additional sensors for HyMeX
with radiative flux measurements, raingauges, a thermosalinograph measuring the near-
surface temperature and salinity, and a thermobathymetric chain giving the ocean
temperature between 5 and 250 m-depth. During SOP1, up to five gliders monitored the
area simultaneously, providing 0−1000 m profiles along repeated transects.
Observations from ships include CTD profiles (up to 200 m-depth) and radiosoundings
from the port-tender Le Provence sent in the Gulf of Lion for 3 IOPs (IOP7, IOP12 and
IOP16). Finally, the freighter Marfret-Niolon that regularly linked Marseille (France) with
Algiers (Algeria), was equipped for HyMeX with the SEOS (Sea Embedded Observation
System; http://dx.doi.org/10.6096/MISTRALS-HYMEX.748) station, measuring air
temperature, relative humidity, pressure, wind and SST. Another sensor provided
measurements of sea temperature at almost 3 m-depth, using a high-quality temperature
probe (TRANSMED data: http://dx.doi.org/10.6096/MISTRALS-HYMEX.973), backed by
a thermosalinograph that also provided in-situ salinity.

### 2.1.2 HyMeX modelling strategy

Despite significant efforts to improve the skill of forecasts, the forecasting accuracy has
been proved still insufficient in terms of amount, timing, and location of heavy



precipitation. The design of the HyMeX modelling strategy considered three key issues
proved to be relevant to reduce modelling uncertainty: (a) to be consistent with the
observation strategy, (b) to integrate numerical models of the atmosphere, ocean, and
land and (c) to include models of the climate system to cover all scales of time and space.
Moreover, through the refinement of model grids and the development of convection-
permitting NWP systems and Regional Climate Models (RCM), significant progress has
been made to improve the simulations of HPEs, the knowledge of the relevant processes
and their interactions across scales, as well as to reduce the large uncertainties on the
future evolution under climate change. The use of finer-scale and coupled models
representing more accurately the atmosphere-ocean-land systems and their
interactions, and/or the detailed validation using the SOP1 measurements allowed the
development of improved parameterizations of physical processes, the design of high-
resolution ensemble modelling approaches with greater number of ensemble members,
and a more sophisticated and efficient use of observations for assimilation purposes.
Profiting from these efforts, the HyMeX community has made relevant advances in
process knowledge and prediction of heavy precipitation. Some of these advances are
discussed and illustrated in the coming sections using the IOP16, which is introduced in
the following.

### 2.2. Illustrative case: IOP 16

The IOP 16 is a well-documented and widely investigated event observed in the period
25-29 October 2012 over the WMed region. IOP 16 was one of the best equipped
observational periods in terms of instrumental coverage during HyMeX-SOP1 (Figure 1).
Most ground-based and air-borne instrumentation were successfully operational,
providing high quality data, with almost all the on-demand SOP1 instruments involved.
Benefiting from this large observational dataset, an extensive number of modelling
activities focused on the IOP 16, with the purpose of investigating different issues related
with the occurrence of heavy precipitation, such as the impact of the turbulence
representation on the sensitivity of the simulated convective systems (Martinet et al.,
2017), the underlying mechanisms of offshore deep convection initiation and
maintenance (Duffourg et al., 2016), some assimilation or pre-assimilation experiments
(Borderies et al., 2019a), the impact of fine-scale air-sea interactions and coupled
processes on heavy precipitation (Rainaud et al., 2017), or novel Large Eddy Simulation
(LES) of a HPE (Nuissier et al., 2020).
This event was associated with a propagating cyclone and was observed in two
dedicated periods: (a) the IOP16a (25−26 October), characterized by heavy





precipitation over CV and LT, when several quasi-stationary MCSs developed, two of
them over the sea, with subsequent heavy precipitation over the French and Italian
coasts on 26 October 2012, and (b) the IOP16b (27−29 October) characterized by heavy
precipitation over CI, NEI, and CO regions.
The IOP16a was driven by the presence of a cyclone moving from the easternmost
Atlantic to the Pyrenees, followed in phase by a cut-off low, associated with upper-level
high potential vorticity values. In the lower troposphere, the cyclone provoked
southwesterly advection of moist and warm air above 20 °C. On the morning of the 26
October the cyclone was centered over the Pyrenees, forming a convergence line
between the southerly flow and the southwesterly colder winds, while over the Tyrrhenian
Sea a southerly moist and warm flow from Tunisia to the Gulf of Genoa established
(Fourrié et al., 2015). During the night from the 25th to the 26th of October and in the
following day several MCSs with quasi-stationary behavior formed within a "comma-
shaped" cloud coverage. First over the sea, between the eastern Spanish coast and the
Balearic Islands (Duffourg et al., 2016), afterwards over the Gulf of Lion inducing large
amounts of precipitation over sea during the morning. The first MCS split in two. One
system (MCS1a) moved towards the south east of the Massif Central, but progressively
decayed producing just orographic rainfall; the second (MCS1b) strengthened and
caused a large precipitation accumulation over the Var region during the afternoon,
nearly 150 mm in 24 h, causing two fatalities in the city of Toulon. A third MCS initiated
at about 06:00 UTC on the 26 October on the Italian coast of Liguria. The MCS
development occurred also offshore Sardinia and Corsica and reached central Italy
during the evening on 26 October, leading to 250 mm daily precipitation on this day over
Liguria-Tuscany, with local flash flooding. On the same day, over the Cévennes-Vivarais
region, daily precipitation reached 170 mm.
During the second period, 27−28 October 2012, the cyclone centre reached the lowest
pressure of 985 hPa over the Alps (Fig. 2), associated with a clear trough in the upper
troposphere and provoking severe northwesterly/northerly winds advecting cold and dry
air over the WMed Sea and inducing large evaporation and ocean cooling and mixing
(Lebeaupin Brossier et al., 2014; Rainaud et al., 2015, 2017; Seyfried et al., 2018). The
relationship between cyclone dynamics and heavy rainfall during IOP16 is discussed in
detail by Flaounas et al. (2015a).

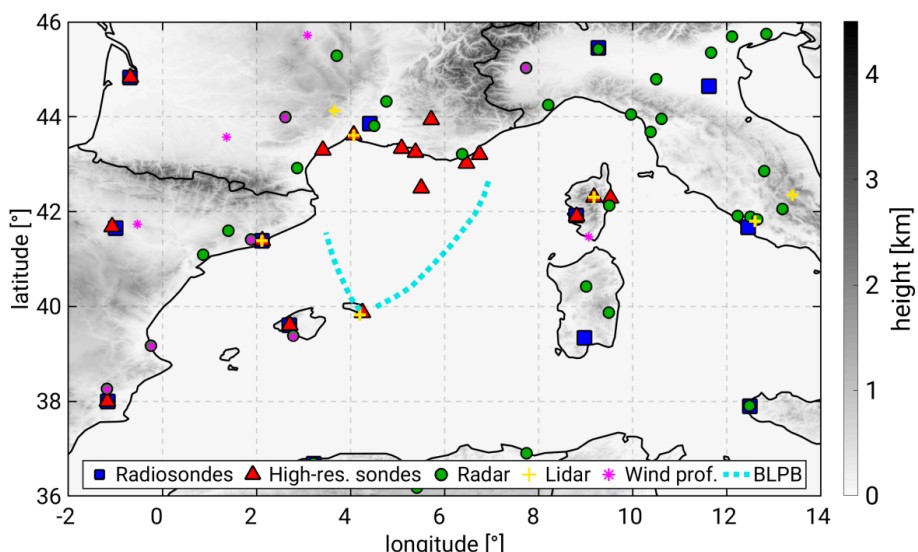

**Figure 1:** Location of selected experimental setup during the IOP16, 25-28 Oct. 2012, including radiosondes and high-resolution sondes, radar, lidar, wind profilers, and Boundary Layer Pressurized Balloons (BLBP). The position of the GPS receivers can be found in Figure 3d.

## 3. Towards improved understanding of the mechanisms leading to heavy precipitation in the Western Mediterranean

### 3.1 Large-scale dynamics and HPE occurrence

Mediterranean cyclogenesis is typically triggered by baroclinic instability because of Rossby wave breaking and the intrusion of upper tropospheric filaments of air masses of high potential vorticity over the Mediterranean (Grams et al., 2011; Raveh-Rubin and Flaounas, 2017). Therefore, most intense Mediterranean cyclones are baroclinic systems with frontal structures and associated airstreams such as dry air intrusions and warm conveyor belts (WCB; Ziv et al., 2009; Flaounas et al., 2015a). The latter corresponds to airstreams that ascend slantwise over the cyclone warm front and are responsible for the characteristic "comma-shaped" cloud coverage of mid-latitude storms. WCBs are associated with stratiform, but also with convective rainfall due to embedded convection within their large-scale ascent branch (Flaounas et al., 2018; Oertel et al., 2019). Such is the case of IOP16, where WCBs and deep convection





coexisted to attribute large amounts of rainfall over the western Mediterranean (Figure
331    2).

Several past studies showed that HP in the Mediterranean basin is intertwined with the
occurrence of cyclones. Scheffknecht et al. (2017) showed that cyclones are present for
all HPEs over the Corsica island when examining the climatology in the period 1985-
2015. Embedded deep convection and WCBs are responsible for the grand majority of
total regional precipitation and its extremes (Jansa et al., 2001; Hawcroft et al., 2012;
Pfahl et al., 2014; Galanaki et al., 2016; Raveh-Rubin and Wernli, 2015). As a token of
cyclones contribution to regional rainfall, Flaounas et al. (2018) showed that the 250
most intense systems of the period 2005-2015 were alone responsible for up to a third
of the total 11-year precipitation, while climate modelling showed that cyclones
contributed from 70% to almost the total of rainfall extremes, depending on the area
(Flaounas et al., 2015b). Such heavy rainfall events are related to water sources from
the Mediterranean Sea, but also from the tropical and extratropical Atlantic Ocean. This
is due to cyclogenesis being preceded by Rossby wave breaking over the Atlantic that
favours the eastward zonal transport of water vapour from oceanic remote areas,
rendering water vapour imports imperative for the formation of heavy rainfall in the
Mediterranean (Duffourg and Ducrocq, 2013; Flaounas et al., 2019).


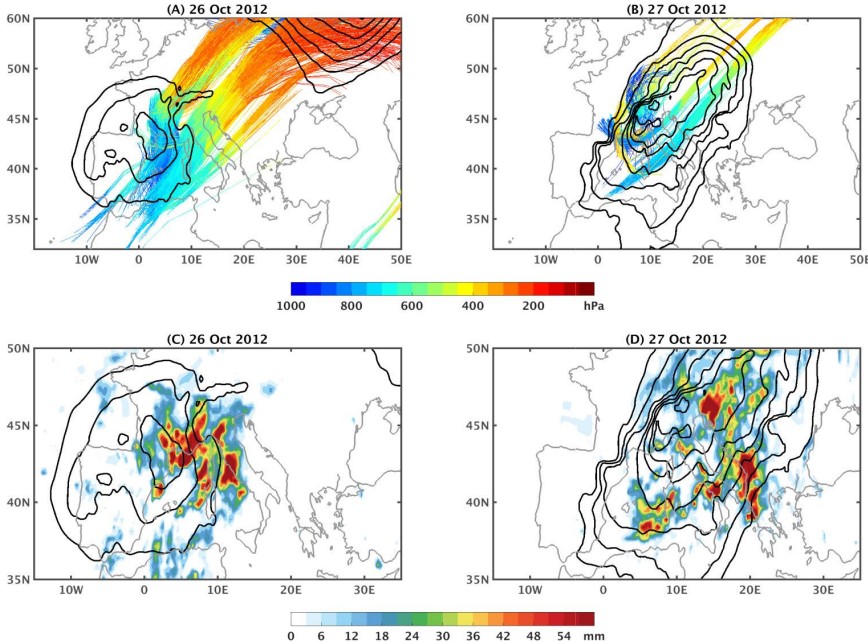


**Figure 2:** (a) Sea level pressure (black contours every 3 hPa, outer contour is set at 1005 hPa). Coloured lines show 48-hour air mass trajectories from ECMWF analyses that correspond to WCBs and where strong ascent takes place at 12:00 UTC, 26 Oct. 2012 (i.e., when cloud ice occurs for the first time in the air masses). Vertical level of the air masses is shown in colour. (b) as in (a), but a strong ascent takes place at 12:00 UTC, 27 Oct. 2012. (c) Daily accumulation of precipitation on 26 Oct. 2012 taken from 3B42 of TRMM (colour). (d) as in (c) for 27 Oct. Datasets and methods are detailed in Flaounas et al. (2015a).

### 3.2 Advances in moist processes understanding

#### 3.2.1 Distribution, origin, and transport of the water vapour supply to HPEs

The relevance of atmospheric water vapour distribution and stratification in the initiation, intensification, and maintenance of HPEs has been extensively demonstrated (e.g., Duffourg et al., 2018; Lee et al., 2018), as well as the role of the Mediterranean Sea as a significant heat and moisture source for HPEs in the WMed area (Duffourg and Ducrocq, 2011; Flaounas et al., 2019). The scarcity of water vapour observations at the mesoscale and smaller scales, as well as the model limitations, for example in relation to the adequate spatial and temporal resolution and/or an accurate representation of the





vertical stratification, hampered progresses in the past. Indeed, our understanding of the
variability of water vapour in relation to convection is still far from being complete. Large
uncertainties remain regarding the origin, pathways, and timescales of transport of the
large amounts of moisture necessary for HPEs in the WMed. The characterization and
better understanding of the water vapour supply to HPEs has been a key aspect of the
HyMex field campaign and subsequent studies. The unprecedented deployment of
instruments during the SOP1 for the monitoring of water vapour dynamics and the
posterior cross-validation studies and synergetic use together with models allowed the
many advances achieved in this period as described in the following.
One of the HyMeX observational highlights has been the dense network of GPS stations,
over one thousand ground-based receivers, providing a reprocessed dataset specially
produced for the HyMeX SOP (Bock et al., 2016). The large extent and high-density
coverage of the reprocessed GPS network allowed a consistent representation of large-
scale features, as well as smaller spatial and temporal scales in agreement with high-
resolution simulations (Bock et al., 2016). Using the reprocessed integrated water vapour
(IWV) GPS data Khodayar et al. (2018) showed that all HPEs within the north-western
Mediterranean form in periods/areas characterized by IWV values in the order of 35-45
mm after an increase of 10-20 mm, being the most intense events those experiencing a
more sudden increase (between 6 to 12 h prior to the event). Bock et al. (2016)
demonstrated that regions prone to HPEs in autumn are characterized by high IWV
variability up to 8 kg/m$^2$.
In addition to the unprecedented (in terms of spatial and temporal coverage) amount of
information provided by the postprocessed GPS network, modelling studies are helpful
for the assessment of potential sources of moisture. Recent advances in this topic
showed that evaporation from the Mediterranean accounts for only about 40% (60%) of
the water vapour feeding the deep convection developed over southeastern France
when cyclonic (anticyclonic) conditions prevails in the days preceeding the event
(Duffourg and Ducrocq, 2013).The Atlantic Ocean (Turato et al., 2004; Winschall et al.,
2011; Duffourg et al., 2018; Flaounas et al., 2019) and tropical Africa (Krichak et al.,
2015; Chazette et al., 2015b; Lee et al., 2016, 2017) have been also suggested as
potential sources of moisture for HPE occurring in the western Mediterranean. The large-
scale uplift of enriched African moisture plume and their role in gradual rain out of the air
parcel over southern Italy during IOP13 were highlighted in a modelling study taking
advantage of stable water isotopes by Lee et al. (2019). Backward trajectory analysis
showed that the large-scale moisture transport takes place during about 3 to 4 days in
the warm sector of front, whereas the surface evaporation over the Mediterranean occurs





shortly in a few hours to 1 day. Associated with extreme precipitation events over Italy,
whether convective or orographic, a recent study by Grazzini et al. (2019) confirmed the
systematic occurrence of anomalously high values of meridional Integrated Vapour
Transport that sometimes occurs in narrow filament shape regions of high integrated
moisture, called atmospheric rivers (Davolio et al., 2020), as during the 2011 Liguria
floods (Rebora et al., 2013) or the last extreme storm in October 2018 (Giovannini et al.,
2021) and October 2020 (Magnusson et al., 2021).
### 3.2.2 Assessment of the variability and vertical distribution of the
### atmospheric water vapour
The variability and vertical distribution of the atmospheric water vapour and their
accurate representation in models have been demonstrated to play a key role for the
timing, location, and intensification of deep convection (e.g., Khodayar et al., 2018), thus
for the simulation of HPEs. They have been further identified as responsible for
inaccuracies in RCMs when compared against convection-permitting NWP models
(Khodayar et al., 2016a). To contribute to the characterization of the water vapour
variability, the ground-based WALI in the Balearic Islands, the airborne water-vapour
differential absorption lidar LEANDRE 2 on board the ATR42 aircraft, and boundary layer
pressurized balloons (BLPB; Doerenbecher et al., 2016) were deployed during the
SOP1. Water Vapour Mixing Ratio (WVMR) profiles were measured with a horizontal
resolution of 1 km (e.g., Flamant et al., 2015; Flaounas et al., 2015a; Chazette et al.,
2015a, 2015b; Di Girolamo et al., 2017; Duffourg et al., 2016; Lee et al., 2016, 2017). In
a multi-instrument and multi-model assessment of atmospheric moisture variability in the
north-western Mediterranean, Chazette et al. (2015b) demonstrated the consistency and
self-coherence of these water vapour data sets during the SOP1 pointing out the strong
need in assimilating high-resolution water-vapour profiles in the lowest layers as those
from lidar instruments. In a multi-scale observational investigation of atmospheric
moisture variability in relation to HPEs formation in the same region, Khodayar et al.
(2018), profited from the synergetic use of the observational datasets demonstrating that
the sampling of spatial inhomogeneities on different scales is crucial for the
understanding of the timing and location of deep convection. Furthermore, focusing on
the complex island of Corsica during SOP1, multiple observations from the mobile
observations platform KITcube (Kalthoff et al., 2013) further demonstrated the benefit of
integrated measurement systems (Adler et al., 2015).
The ground-based lidar WALI was useful in capturing the moist and deep boundary
layers with updrafts reaching up to 2 km in pre-convective environments leading to HPEs,





contrary to the dry, shallow boundary layers everywhere else (Khodayar et al., 2018). In
Chazzete et al. (2015a), the ground-based lidar WALI, additionally captured the
increasing moistening of the free troposphere, up to 5 km, prior and in relation to the
MCS formation. Furthermore, the specific humidity observations from BLPB and aircraft
flights captured spatial inhomogeneities in the lower boundary layer up to 4 g/kg in less
than 100 km, which were shown to determine the location of convection initiation
(Khodayar et al. 2018).
Figure 3 illustrates for the IOP16 the complex moisture flow that fed the convective
systems, which was effectively monitored by the variety of water vapour profiling sensors
involved in combination with backward and forward trajectory analyses from a
Lagrangian model (NOAA HYSPLIT Lagrangian trajectory model; Draxler and Hess,
1998; Rolph et al., 2017; Stein et al., 2015) and the information derived from the GPS
network.
In Figure 3a, the spatial distribution of the 24 h-averaged GPS-derived IWV on 25 and
26 October 2012 shows initially higher atmospheric moisture content in the western area,
where convection initiation takes place, whereas on the next period the humid air mass
has advanced eastwards. The water vapor mixing ratio as measured by WALI in
Ciutadella (Menorca) (Figure3b) reveals the presence of two distinct humid layers in the
time interval 18:00−00:00 UTC on 25 October 2012: a surface layer extending up to
about 0.6 km, with mixing ratio values up to $\sim$12 g kg$^{-1}$, and an elevated layer extending
from 1.1 to 2.5 km, with mixing ratio values up to 6−7 g kg$^{-1}$. The 24 h forward trajectory
analysis starting in Ciutadella at 21:00 UTC on 25 October 2012 at the altitudes of the
observed humidity layers (Figure 3c) shows the northward movement of the surface
humid layer, while air masses within the elevated humidity layer moved north-eastward.
The latter are plausibly related to the inflow branch of WCBs, i.e., the blue part of the air
mass trajectories in Figure 2. The surface humid layer overpassed Candillargues about
15−21 hours later, as proved by the mixing ratio profile measurements carried out by
BASIL and illustrated in Figure 3e, possibly feeding the MCS forming close to the cyclone
centre over the Cévennes-Vivarais region in the morning of 25 October. Indeed, the
water vapour profile shown in Figure 3e is consistent with the one provided by LEANDRE
2, on board the ATR42 aircraft that flew over the WMed (Figure 12 in Flaounas et al.,
2015a). The 24 h forward trajectory analysis also reveals that air masses within the
elevated humid layer overpassed the Gulf of Lion, possibly ending up with feeding the
offshore MCS system. Figure 3d additionally illustrates the 200-h back-trajectory
analysis ending in Minorca at 21:00 UTC on 25 September 2012 at the altitudes (500 m





and 2000 m) where the two humid layers were observed, revealing that air masses within
the surface humidity layer originated over the tropical Atlantic Ocean approximately 8
days earlier and overpassed Morocco and Southern Spain, slowly subsiding (in the last
72-96 hours before the formation of them MCS) upon reaching the Mediterranean basin
from an altitude of 1000 m down to 500 m, whereas the air masses within the elevated
humid layer originated over Central Africa (Northern Mali) approximately 3 days earlier
and transited over Mauritania and Morocco before reaching the Balearic Islands.

482        (a)                           (b)

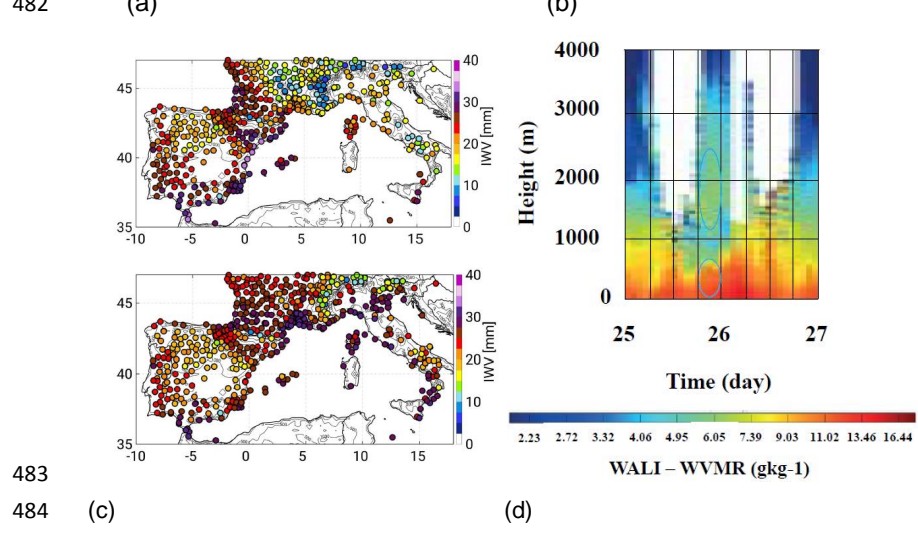


484        (c)                           (d)





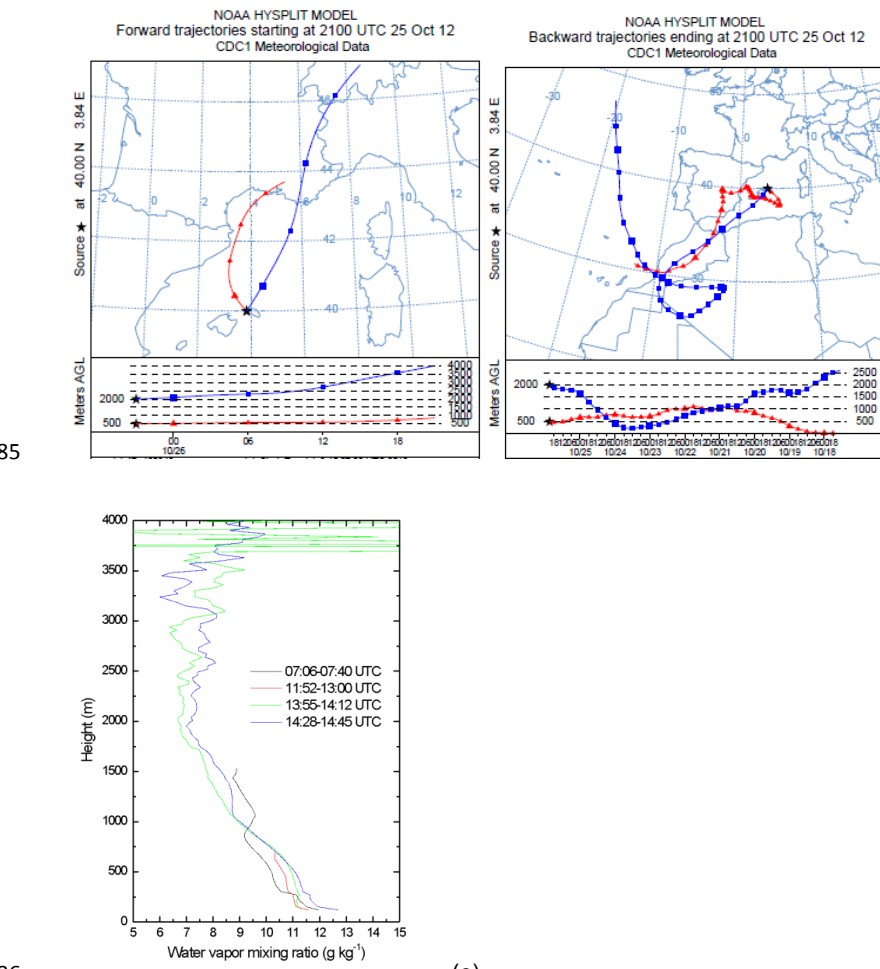


486                                                                (e)

**Figure 3:** (a) Spatial distribution of 24 h-averaged GPS-derived IWV (mm) on   25 (top), and 26 (bottom) Oct. 2012. (b) Time evolution of the water vapor mixing ratio (g kg$^{-1}$) as measured by the ground-based lidar WALI in Menorca over the 48-h period from 00:00 UTC, 25 Oct. 2012 to 00:00 UTC, 27 Oct. 2012. (c) 24-h forward trajectory analysis from HYSPLIT starting in Ciutadella (Menorca) at 21:00 UTC, 25 Oct. 2012 and ending at 21:00 UTC, 26 Oct. 2012 and (d) 200-h back-trajectory analysis from HYSPLIT ending in Ciutadella (Menorca) at 21:00 UTC, 25 Oct. 2012. (e) Vertical profiles of the water vapor mixing ratio (g kg$^{-1}$) as measured by the ground-based lidar BASIL in Candillargues at different times on 26 Oct. 2012.

**3.3 Low-level dynamical processes**



Once the synoptic setting becomes favourable for heavy precipitation in the WMed , with
an upper-level trough slowly evolving eastward while deepening over the basin, the
mesoscale organization and the thermodynamic characteristics of the low-level flow
determines the occurrence, intensity and location of heavy precipitation. Most of the
severe rainfall events that occurred during the SOP1 field campaign can be connected
or at least interpreted in the framework of recent theoretical results concerning moist
orographic convection (Miglietta and Rotunno, 2014; Kirshbaum et al., 2018). However,
their in-deep analysis has revealed a greater complexity of real meteorological situations,
due to non-stationarity, to the complexity of the real 3D orography and vertical profiles,
and especially to the interaction among small-scale processes, which are not entirely
accounted for in controlled-environment numerical experiments. One of the merits of
HyMeX has been to provide evidence of the theoretical results and to enrich our
knowledge on genesis and evolution of convection in a complex topography environment
through a plethora of modelling simulations and tools, and advanced instrument
observations.
Being heavy orographic precipitation in stable and neutral atmospheric conditions
already investigated in past experiments (e.g., MAP-Mesoscale Alpine Programme,
Bougeault et al., 2001) and well understood, the focus of HyMeX was on the
development of quasi-stationary MCSs, well known responsible of recent HPE and floods
in the area (Nuissier et al., 2008; Buzzi et al., 2014; Romero et al., 2014 among others).
These systems are characterized by "back-building" processes that force the continuous
redevelopment of deep convective cells over the same area producing severe and
persistent rainfall (Schumacher and Johnson, 2005; Ducrocq et al., 2008; Duffourg et al.,
2018; Lee et al., 2018). The multicell MCSs resulting from this retrograde regeneration
assume a typical V-shaped pattern in radar and satellite images. In this context,
conditionally unstable marine flow directed towards the coastal mountainous regions and
extracting energy from the sea surface has been pointed out as a common feature of all
the events. However, different convection-triggering mechanisms have been identified
and highlighted.

### 526 3.3.1 *Convection-triggering mechanisms*

Low-level convergence over the sea can initiate convection even far from the coast and
usually it is produced by the large scale forcing. During IOP16 (Duffourg et al., 2016) the
cyclonic circulation around a shallow low-pressure system was responsible for low-level
convergence against the south-easterly flow, between Balearic Island and the Gulf of
Lion (Figure 4a). Lee et al. (2016) revealed the key role during IOP13 of an approaching



cold front in modifying the low-level circulation over the Tyrrhenian Sea, establishing
favourable dynamical conditions for convection initiation. Even for the IOP8, the low-level
convergence that first triggered convection south of the Iberian Peninsula was ascribed
to the large-scale setting (Röhner et al., 2016; Khodayar et al., 2015), even if orographic
effects were essential to enhance mesoscale uplift over land during the mature phase of
the convective system.
In fact, due to the peculiar topographic characteristics of the basin, in most of the events
it is the interaction with the orography that triggers and eventually maintains convection,
since it does provoke not only the direct lifting, but can also produce the convergence
required to initiate vertical motions. Several numerical experiments (Barthlott and
Davolio, 2016) clearly showed the effects of Corsica and Sardinia on the downstream
low-level wind as well as on temperature and moisture distribution. In particular, the
deflection of the westerly/south-westerly flow due to the complex orography of the islands
was identified as a key mechanism for the organization of heavy precipitation along the
western Italian coast, since it determined small-scale complex patterns of low-level
convergence over the sea in the lee of Corsica, where convection was triggered (Figure
4b). Moreover, the interaction between sea breezes and drainage winds induced by
mountainous islands like Corsica or Sardinia (Barthlott and Kirshbaum, 2013; Barthlott
et al., 2016) impacts on the development of deep convection both offshore and anchored
to topographic features. Also, the flow splitting around Corsica Island can be a key
mechanism producing a lee-side convergence line where a severe and stationary
convective system develops (Scheffknecht et al., 2016).
Interestingly, the study of Lee et al. (2017) clearly indicated that neither an offshore
convergence line nor the orographic uplift alone would have been enough to allow the
development of the intense MCS that affected the Ebro River valley during IOP 15
(Figure 4c). It was their interplay that produced deep convection, together with the
simultaneous presence of flow channelled by the local orography and converging with
the low-level marine inflow. This represents a clear example of the complex interaction
among processes that HyMeX was able to highlight.
Low-level convergence induced by blocking effect of mountain chains on the impinging
flow is another frequent lifting mechanism upstream of the orography. Well before the
HyMeX SOP, it was demonstrated that flow blocking in front of Massif Central and the
enhanced convergence due to deviation of southerly flow around the Alps (Figure 4d)
(Ducrocq et al., 2008; Davolio et al., 2009), were responsible for several HPE over
southeastern France, affecting areas well upstream of the main orographic reliefs. SOP





related studies identified a similar low-level flow characteristic, associated with heavy
rainfall over north eastern Italy. In both the analysed events, occurred during IOP2b
(Manzato et al., 2015; Miglietta et al., 2016) and IOP 18 (Davolio et al., 2016), the
blocking of southerly low-level marine inflow in the form of a north-easterly barrier wind
in front of the Alps, produced strong and localized convergence, favouring convection
triggering (Figure 4e). Through additional modelling investigation of similar events in the
past, this was recognised as a typical mechanism for deep convection (even supercell)
development over the area.
The importance of orographic interaction has been revealed also for the development of
lee-side convection. Pichelli et al. (2017) through a number of numerical simulations of
IOP6 illustrated the complex and delicate equilibrium between competing processes
(orographically induced subsidence on the lee side and frontal uplift) that determined the
evolution of a squall line over the Po Valley, in the lee side of a mountain range
(Apennines) with respect of the main southerly flow feeding the precipitation.

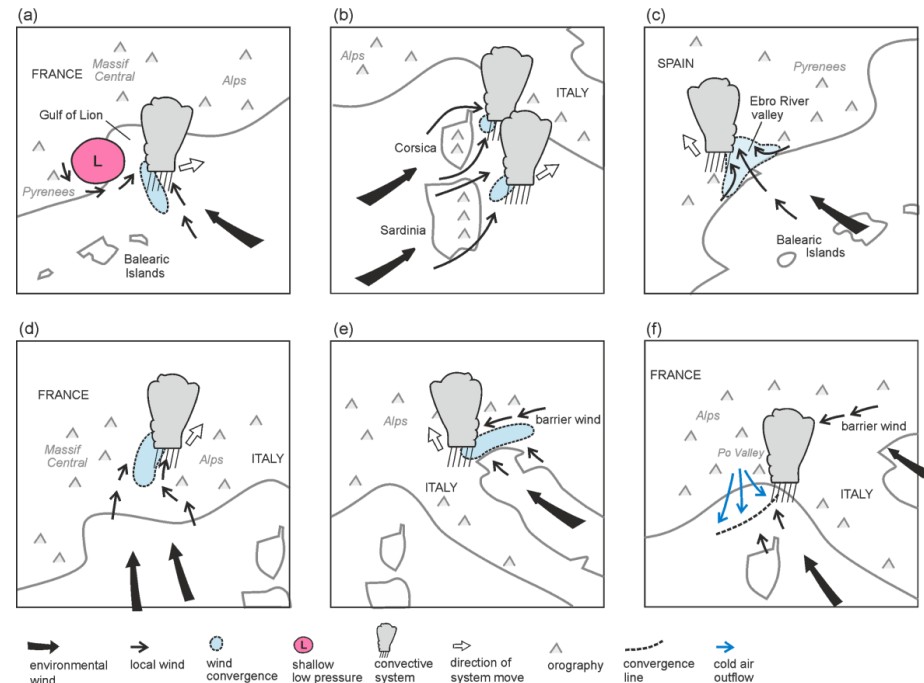


**Figure 4:** Conceptual illustrations of key convection-triggering mechanisms in the north-
western Mediterranean basin. Coast lines are depicted by grey solid lines.





### 3.3.2 Cold pools

The detailed observational and modelling analysis of IOP13 revealed that, as expected, also the direct orographic uplift can trigger convection close to the coastal slopes. However, thanks to detailed observations and modelling simulations of the precipitation system and of the upstream environment, Duffourg et al. (2018) were able to provide a thorough description of the mechanisms that maintained the MCS while slowly moving offshore. In fact, the formation of an evaporative cold pool under the precipitating cells generated down-valley flows that slowly shifted the location of the back building convective cells from the mountain to the coast and over the sea.

In this regard, it was emphasized (Lee et al., 2018) that the moisture vertical distribution in the lower troposphere can modulate the intensity of the cold pool and thus to control location and amount of heavy precipitation associated with the MCS. In several other events, the leading edge of a cold pool, formed by evaporative cooling under the precipitating cells, was able to trigger convection by lifting the impinging ambient low-level flow. As suggested by idealized experiments of conditionally unstable flow over a mountain ridge (Bresson et al., 2012; Miglietta and Rotunno, 2009), the stationarity of the MCS or its upstream propagation away from the orographic barrier is determined by the intensity of the ambient flow. In this context, the vertical structure of the lower troposphere, in terms of moisture content and wind intensity, represents an important factor since it modulates the evaporation potential and thus the formation and intensity of the cold pool.

However, the presence of cold and dense air acting as a virtual mountain with respect to the impinging warm and moist flow can be due to different processes besides evaporative cooling. In the analysis of IOP8, Bouin et al. (2017) identified cold and moist air masses transported from the Gulf of Lion by the low-level jet. Despite their moisture content, these air masses were cold and dense enough so that their accumulation on the foothills of the relief contributed to initiating a cold pool. Once the MCS was triggered, rain evaporation in the subsaturated mid-level layer resulted in downdraughts that further intensified the cold pool, favouring the regeneration of the precipitation system. Finally, investigation of heavy precipitation over Liguria in IOP16 as well as in previous dramatic HPE, undertaken within the HyMeX framework, provided a clear picture of the mechanisms responsible for recent and recurrent disastrous floods along the Ligurian Sea coast. Several studies (Buzzi et al., 2014; Fiori et al., 2017 among others) pinpointed the role of the cold air outflow from the Po Valley, across the Apennine gaps, which propagate as a density current to the Ligurian Sea, where it determined a sharp



mesoscale convergence line (sketched in Figure 4f). Along such a convergence line, the
lifting of southerly moisture laden flow produced the onset of the severe convection.
Interestingly, the cold flow over the sea appeared to be induced by an easterly inflow into
the Po Valley from the Adriatic side, possibly due to a barrier wind effects over north-
eastern Alps as previously described. As observed in many other cases (e.g., Duffourg
et al., 2016) although the V-shape structure seems anchored over the sea, a few tens of
kilometres offshore, intense convective cells are continuously advected inland where HP
occurs. Finally, Duffourg et al. (2016) also highlighted an interesting feedback process
of convection to the environment that, through small-scale perturbations of the low-level
circulation around the cold pool, focussed and reinforced the local moisture convergence
feeding the convective updraft.

### 3.4 Impacts of the land and the sea surfaces

#### 3.4.1   Land conditions and feedback to the atmosphere

Land conditions and feedbacks between the land surface and the atmosphere play a role
in determining the response of the Earth system to climate change, particularly in the
Mediterranean region, which is a transitional zone between dry and wet climates. Indeed,
enhanced land–atmosphere feedbacks are expected in a warming climate, and their
understanding and simulation are challenging, but fundamental to further improve our
knowledge about future climate and their interactions with the other components of the
climate system. Despite its relevance, the modelling of land-atmosphere feedbacks still
suffers for relevant uncertainty owing to inaccurate initialization, and/or model physics,
misspecified parameters, etc… Helgert and Khodayar (2020) showed that an
improvement of the soil-atmosphere interactions and subsequent HP modelling is
observed using an enhanced initialization with remote sensed 1 km SM information.
Khodayar and Helgert (2021) additionally investigated the response of the western
Mediterranean HP to extreme SM conditions showing that changes in the initial scenarios
impact the mean, but also the extremes of precipitation. Regional projections of
precipitation under the RCP4.5 and RCP8.5 scenario have shown to be considerably
modified when SM is used as predictor (Hertig et al., 2018). Therefore, a better
knowledge and representation of soil conditions and evolution have to be considered for
HP understanding and modelling.

#### 3.4.2   Air-sea interactions and coupling

The Mediterranean Sea and the atmospheric boundary layer (ABL) continuously
exchange momentum, heat, and freshwater. These exchanges, related to the turbulent
fluxes, are controlled by the gradients of temperature, humidity, and velocity at the air-
sea interface. Rainaud et al. (2015) showed that although moderate air-sea fluxes were



observed during the HPEs of SOP1, large air-sea exchanges in the Gulf of Lion and the
Balearic, Ligurian and Tyrrhenian Seas can be correlated to the occurrence of a HPE.
The SST strongly influences the low-level flow stability and dynamics through heating,
moistening and downward momentum mixing (Stocchi and Davolio, 2017; Meroni et al.,
2018a). SST is indeed a key parameter for evaporation (Figure 5a) and its influence on
HPEs in terms of convection triggering, intensity, and location, has been extensively
investigated with several numerical studies (e.g., Strajnar et al., 2019; Senatore et al.,
2020b, for some of the most recent). Generally, these studies highlight that the SST
values strongly and directly modify the low-level atmospheric stability, which first impact
the intensity of convection and precipitation, with the most intense rainfall associated
with warmer sea surface. The location and stationarity of heavy precipitating systems
are also modified, with an acceleration of the low-level wind velocity over warmer sea,
but also by the fine-scale SST horizontal patterns with eddies and marked fronts in the
Mediterranean (as explicitly simulated in the coupled forecast of Rainaud et al. (2017)
for IOP16a shown in Figure 5a) that can significantly change the flow dynamics
interacting with orography (Davolio et al., 2017) or displace the moisture convergence at
sea (Rainaud et al., 2017; Meroni et al., 2018a).
During intense meteorological events in the Mediterranean such as HPEs, significant
modifications of the ocean mixed layer (OML) can occur, even on short timescales of
only several hours (Lebeaupin Brossier et al., 2014), and can significantly impact the
exchanges with the ABL. Berthou et al. (2016) showed that IOP16a was likely sensitive
to SST changes upstream related to OML changes and sub-monthly air-sea coupling.
The ocean vertical stratification is also a characteristic which has to be accounted for, as
sea surface cooling during HPEs is largely controlled by the entrainment of deeper and
colder water in the OML. The study of Meroni et al. (2018b) using coupled experiments
with idealized ocean conditions highlights that the cooling is more pronounced with a
shallow, strongly stratified OML, leading to lower air-sea fluxes, less air instability and
finally a reduction of the total amount of simulated precipitation.
The use of ocean-atmosphere coupled systems enables us to consider the ocean 3D
structure and its interactive and consistent evolution. Such modeling systems were
specifically developed in the framework of HyMeX to improve the realism of weather
forecasts for sea surface and atmospheric low levels, and innovatively evaluated thanks
to the multi-compartments' observational dataset. For IOP16a, Rainaud et al. (2017)
showed that, due to mixing and heat loss, the progressively lower SST in the coupled
model induces lower heat fluxes (-10% to -20% of evaporation), local differences in the





low-level environment (stability) and cyclonic circulation, with small impacts on the
convection organization (convergence) and intensity.
Waves also impact the atmospheric low levels, by increasing the surface roughness and
the momentum flux and act as a drag for the low-level upstream flow. For HPEs, in forced
and coupled simulations, this slow-down results in modification of HP location due for
instance to changes in the low-level flow, convergence line or cold pool motion over the
sea (Bouin et al., 2017; Sauvage et al., 2020). The study of Thévenot et al. (2015)
highlighted this waves impact in the IOP16a case, with a better representation of the low-
level moist jet feeding the MCSs and of the simulated precipitation when sea state is
considered in the bulk formulae (Figure 5b).

(a)

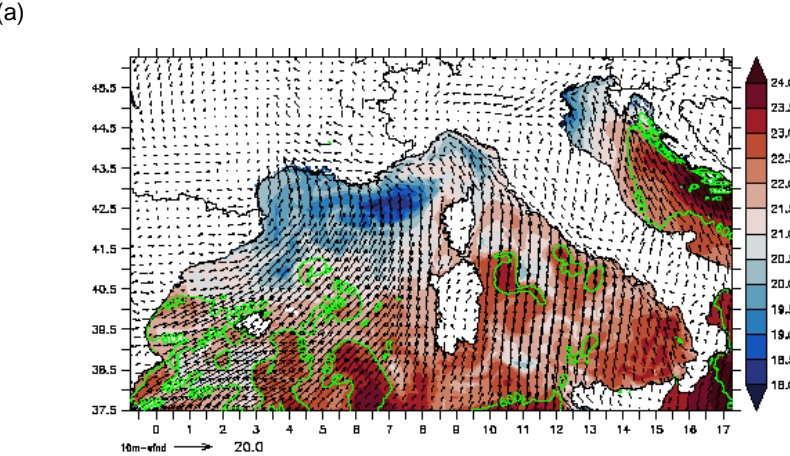

(b)

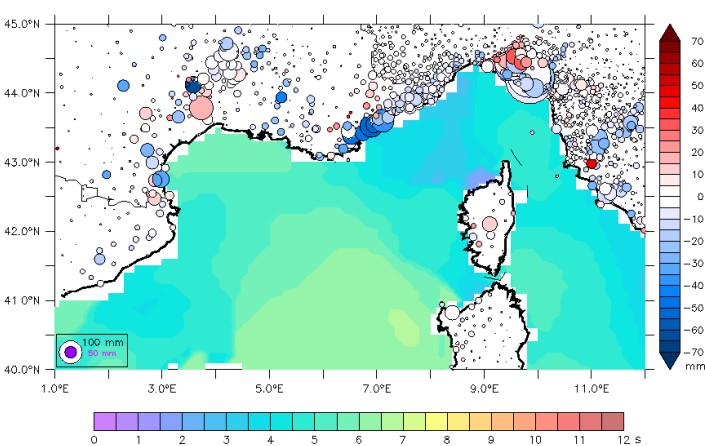



**Figure 5**: (a) Daily mean SST (colours, °C), 10m-wind (arrows, m s$^{-1}$) and surface evaporation (green contours for values above 600 kg m$^{-2}$) for 26 Oct. 2012 (IOP16a) from the AROME-NEMO WMED coupled experiment (CPLOA) of Rainaud et al. (2017). (b) Peak period of waves (color, s) at 00:00 UTC, 26 Oct. 2012 considering in Thévenot et al. (2016) and bias modification (circles, mm) for 24h-rainfall accumulation against rain-gauges data, comparing MESO-NH simulations with (WAM) and without (NOWAV) sea state impact (blue for an improvement in WAM). The size of the circles indicates the NOWAV bias (absolute value, in mm).



### 3.5 Microphysics

Many advances in the understanding and knowledge of cloud composition and microphysical processes in Mediterranean convective systems were attained in the framework of HyMeX, thanks to the large number of observations used in process, modeling and/or data assimilation studies.
Among them, a large number of available disdrometers and MRRs were used to improve the quality of observations (Raupach and Berne, 2016 and Adirosi et al., 2016) and the characterization of the raindrop's PSD (Adirosi et al., 2014, Adirosi et al., 2015, Schleiss and Smith, 2015), including its very small-scale variability (Gires et al., 2015).
Based on rain gauge observations over a long period encompassing the HyMeX experiment, Molinié et al. (2012) studied the rainfall regime in a mountainous Mediterranean region, in southeastern France. They found that rainfall intermittency, both at the monthly and daily scales, is well correlated to the rain gauge altitude, which is also linked to rainfall intensity. Zwiebel et al. (2015) and Hachani et al. (2017) also found that several factors (altitude, season, weather type, among others) influence both the rainfall characteristics at the ground and the relationship between rainfall rate and the reflectivity factor.

720 Other studies focused on the use of radar data to investigate the cloud composition. Grazioli et al. (2015) proposed a hydrometeor classification algorithm using an X-Band radar (deployed in Ardèche during HyMeX SOP1). Ribaud et al. (2015) also developed a hydrometeor classification algorithm using dual-polarimetric radars and produced 3D hydrometeor fields when several radars were available. Using this classification, they also identified a link between cloud characteristics and lightning propagation (Ribaud et al., 2016).

HyMeX microphysical observations have also led to improvements in model physics and parameterizations. Fresnay et al. (2012) first demonstrated the sensitivity of





Mediterranean HPEs simulations to the cloud parameterization. Using several
observations from HyMeX SOP1, Taufour et al. (2018) showed that the 2-moment
scheme LIMA (Vié et al., 2016) provides a more realistic cloud representation than the
1-moment scheme ICE3. This is shown in Figure 6 comparing observed and simulated
RASTA reflectivities. The shape of the reflectivity distribution is better represented by
LIMA than ICE3, especially in the melting region. Furthermore, they proposed a revision
of the scheme LIMA based on the disdrometer rain PSD observations.
The aerosol-cloud interactions were also found to have a strong impact on convective
systems and rainfall characteristics (Kagkara et al., 2020), and the best simulation results
with the 2-moment scheme LIMA are obtained when using a realistic aerosol population
from the MACC analyses validated against ATR42 observations (Taufour et al., 2018).
Eventually, some studies prepared the future assimilation of cloud data. Augros (2015)
implemented the assimilation of dual-polarization radar data in the French operational
AROME model. Borderies et al. (2019a, 2019b) proposed a method to assimilate
airborne RASTA reflectivities and Doppler winds, meanwhile releasing an improved
version of the RASTA simulator for use in mesoscale models.

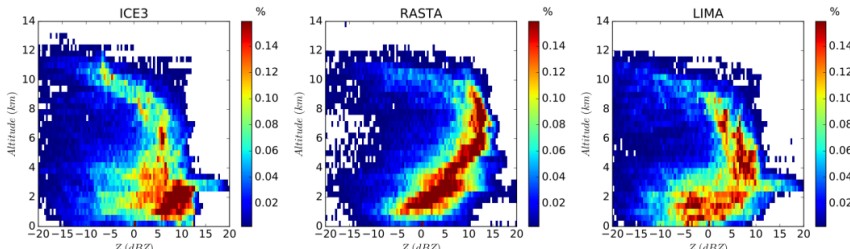


**Figure 6:** Comparison of observed and simulated RASTA reflectivities, merging data
from IOPs 6 and 16a (From Taufour et al. 2016).

**4.  Improving Heavy precipitation modelling across scales**
**4.1 Increasing model resolution simulations**
Idealized simulations of deep moist convection at kilometric scales (grid spacing: 4 km,
2 km, 1 km and 500 m) showed that the accumulated rainfall and corresponding surface
area, as well as the area covered by the updrafts, increase with increasing resolution. At
4 km horizontal resolution, deep convection is under-resolved, and differences are larger
between 1 km and 500 m horizontal resolution simulations than between 2 and 1 km,



suggesting the beginning of convergence at 500 m (Verrelle et al., 2014). Bassi (2014)
analysed several IOPs over LT target area performing numerical simulations at different
grid-spacings between 3 and 1 km, and with different resolutions of the orography
representation, showing that both aspects equally contributed to improve the quantitative
precipitation forecast (QPF). In fact, the higher model resolution allowed a better
description of the structure, vertical motions, and dynamical mechanisms of the
convective system, whereas accurate orography was required to correctly simulate the
propagation of the density current along the Apennine slopes, and thus the precise
location of the convergence line that triggered the MCS. In terms of microphysics
parameters, discrepancies between models and observations could be attributed to the
implementation of one-moment microphysics scheme and to the coarse resolutions,
hence, there is a need for grid spacing finer than 2.5 km (Augros et al., 2015).
The increase in horizontal resolution is therefore a great improvement but it additionally
poses challenges for the model physics since some parameterization schemes may
become inappropriate. This is the case, for example, of the turbulence parameterization
in the "grey zone" between 1 km and 100 m horizontal grid spacing (Wyngaard and Coté,
1971), or of one-moment microphysical schemes where the overestimation of
reflectivities at high altitude due to graupel is a known limitation (Varble et al., 2011).
Hectometric-scale simulations of a Mediterranean HP event at 150 m by Nuissier et al.
(2020) were able to capture features regarding convective organization within the
converging low-level flow, which are out of range of models with kilometric horizontal
resolutions. However, the comparison of the large-eddy simulation (LES) with a
reference simulation performed with a 450 m grid spacing in the heart of the so-called
"grey zone" of turbulence modelling shows that the increase in resolution does not
significantly reduce deficiencies of the simulation, being this fact more related to an issue
of initial and lateral boundary conditions.
**4.2 New generation of high-resolution convection permitting simulations**
**and improvement of RCMs**
One of the most remarkable advances in the last years with significant implications for
HP simulation has been the development of the new generation of high-resolution
convection-permitting models (CPMs). This development has been extensively fostered
and exploited in HyMeX related activities and studies and represented one of the main
innovations contributing to advance knowledge in HP occurrence. Kilometric grid spacing
has become achievable with the increasing availability of computational resources. As
the horizontal resolution approaches 1 km, parameterization of deep convection is no





longer needed since much of the convective motion is explicitly resolved. It has been
demonstrated that the reduction of the grid spacing leads to a weaker overestimation in
height and size of the convective cells (Caine et al., 2013), together with a more accurate
representation of the timing and location of convection (Clark et al., 2016).
The benefit of higher horizontal resolution of CPMs can also propagate along the
forecasting chain to hydrological predictions. Simulating the catastrophic Liguria floods
of 2011, Davolio et al. (2015) demonstrated that the finer grid resolution resulted in better
QPF because of a more accurate description of the MCS and of its interaction with the
orography, and this improvement was confirmed also in terms of discharge forecasts.
In a seamless weather-climate multi-model intercomparison, Khodayar et al. (2016a)
showed that despite differences in their representation of a HPE, CPMs represented
more accurately the short-intense convective events, whereas the convection-
parameterized models produce a large number of weak and long-lasting events and
RCMs produce notably lower precipitation amounts and hourly intensities. Figure 7
shows an example of how finer resolution simulations better represent convergence over
the sea, where warm and moist air is transported by a low-level jet towards the French
coast. The higher resolution enhances the humidity convergence areas over the sea,
which appear located further upstream, as well as the associated triggering of
convection. Furthermore, the added value of convection-permitting with respect to RCMs
has also been demonstrated in the north-western Mediterranean basin (e.g., Berthou et
al., 2018; Coppola et al., 2018; Fumière et al., 2019). Berthou et al. (2018) showed that
convection permitting RCM simulations (about 2.5 km grid spacing) better represented
HPE in southern France in terms of daily precipitation than their convection-
parameterized counterparts (about 12.5 km grid spacing). It was also shown the added
value for the simulation of hourly rainfall over the United Kingdom, Switzerland, and
Germany. Coppola et al. (2018), in a multi-model study, proved the ability of high-
resolution CP-RCMs to reproduce three events of HP, one in summer over Austria, one
in fall associated with a major Foehn event over the Swiss Alps and another intense fall
event along the Mediterranean coast. In a dedicated study of Mediterranean HPEs in fall
on an hourly time scale, Fumiere et al. (2019) demonstrated that high resolution allows,
(a) the improved representation of the spatial pattern of fall precipitation climatology, (b)
the improvement of the localization and intensity of extreme rainfall on a daily and hourly
time scales, and (c) the ability to simulate intense rainfall on lowlands.



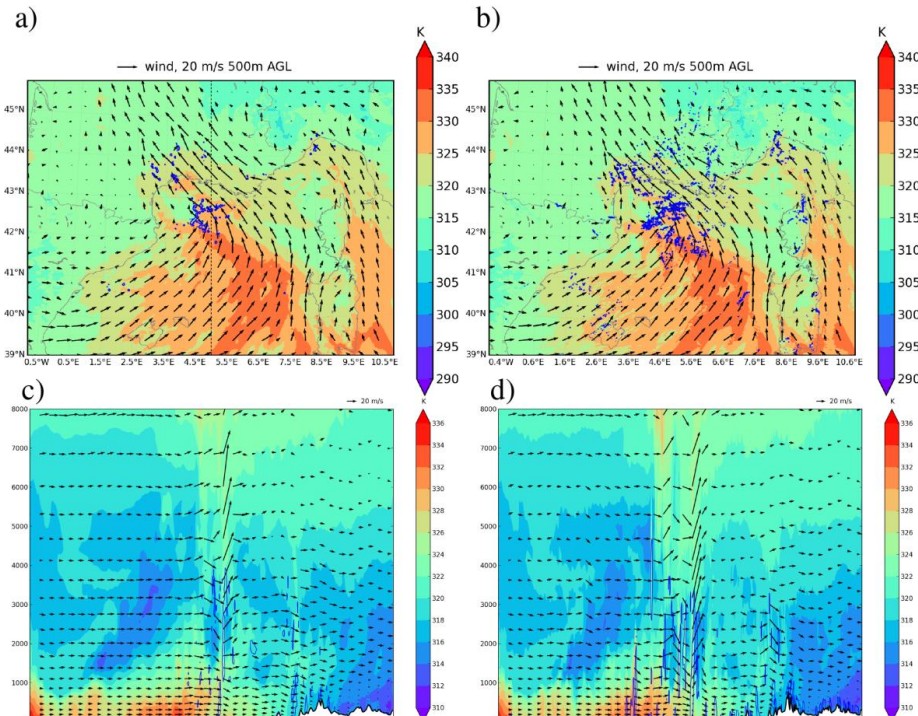


**Figure 7**: IOP16: 0930 UTC,26 Oct. 2012. Horizontal cross sections at 500m AGL and vertical cross sections along a South-North line (shown in a) of equivalent potential temperature (K, in colour scale), and wind vectors (m s$^{-1}$, black arrows) for 2-km (left panels) and 500-m (right panels) resolution runs. The blue lines represent the humidity convergence (-0.1 kg s$^{-1}$ m$^{-3}$ for vertical cross sections, and -0.02 kg s$^{-1}$ m$^{-2}$ integrated value over the layer between the ground and 3000 meters for horizontal cross sections).

## 4.2 Improvement of parameterization schemes

Recent studies have shown that the simulation of convective systems is very sensitive to model parameterizations. For the IOP16a, Thévenot et al. (2015) showed that taking sea state into account in the turbulent air-sea exchanges can modify the low-level dynamics of the atmosphere and the precipitation location. However, the relationship of Oost et al. (2002) used in this study to compute the roughness length is known to overestimate the turbulent fluxes in strong wind regimes. New formulation of sea surface turbulent fluxes parameterization is under development and currently tested to better represent the wind-sea (i.e., the younger waves locally generated by wind) impact and related variability. The preliminary results when applied to HPE forecasts confirm the



significant slow-down of the upstream low-level flow with displacement of convergence
over the sea and show minor changes in the heat and moisture fluxes (Sauvage et al.,
2020). Further developments are planned concerning sea surface fluxes computation,
including notably the impact of sea spray on moisture and of the swell (i.e., the oldest
non-local waves).
In Rainaud et al. (2015), a change in the SST or the coupling of atmospheric and oceanic
models is found to have a large impact on the simulated precipitation amount over land.
Martinet et al. (2017) investigated the sensitivity of simulated HP at a sub-kilometric scale
(500 m) to the turbulence parameterization (i.e., Deardorff or Bougeault-Lacarrère)
showing that the simulated environment and convective processes are highly sensitive
to the formulation of the mixing-length. Convective systems are more intense in
association to larger moisture advection, higher hydrometeor contents and marked low-
level cold pools with weaker mixing lengths, since in this case the subgrid TKE is weaker,
and winds are increased to balance this effect.
Moreover, Verrelle et al. (2014) found insufficient turbulent mixing inside convective
clouds, more pronounced at kilometer resolution with weak thermal production,
underlying a lack of entrainment in convective clouds at intermediate range (between
500 m and 2 km horizontal resolution). By using LES of deep convection, Verrelle et al.
(2017) and Strauss et al. (2019) showed that the commonly used eddy-diffusivity
turbulence scheme (K-gradient formulation) underestimated the thermal production of
subgrid TKE and did not enable the nonlocal turbulence due to counter-gradient
structures to be reproduced. These two studies also found that the approach proposed
by Moeng (2010), parameterizing the subgrid vertical thermodynamical fluxes in terms
of horizontal gradients of resolved variables (H-gradient approach), reproduced these
characteristics, and limited the overestimation of vertical velocity. This new approach has
also been assessed using Meso-NH simulations at kilometer-scale resolutions for real
cases of deep convection on two HyMeX IOPs (IOP6 and IOP16a) (Ricard et al., 2021).
The new scheme enhances the subgrid thermal production of turbulence with a better
representation of counter-gradient areas and reduces the vertical velocity inside the
clouds (Figure 8). The enhanced turbulent mixing modifies the entrainment and
detrainment rates and produces more developed anvils with increased values of ice and
snow, which are more realistic. It also affects the cold pool under the convective cells.



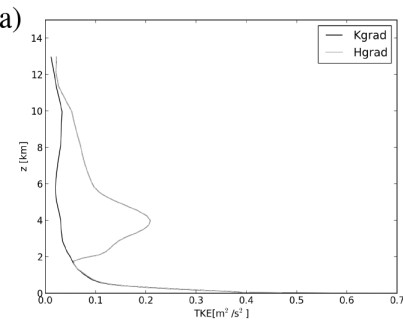 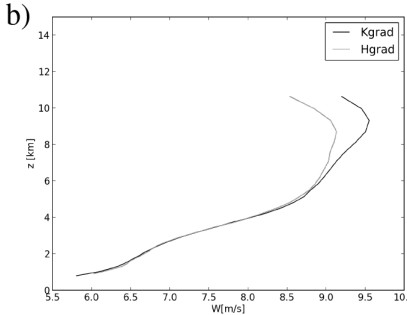


**Figure 8:** Mean vertical profiles inside the clouds of (a) subgrid TKE (m² s⁻²) and (b) vertical velocity (m s⁻¹) during IOP16 (between 00:00 UTC, 26 Oct and 00:00 UTC, 27 Oct. 2012) for 2-km horizontal resolution Meso-NH simulations using K-gradient formulation (black line) and H-gradient formulation (grey line) for the vertical turbulent fluxes of heat and moisture.


### 4.3 Data Assimilation

One of the HyMeX goals was to improve or develop research- as well as operational-
oriented atmospheric data assimilation systems and methods. Emphasis has been put
on progresses in the processing of observations currently available in data assimilation
systems and on the assimilation of new observation types, especially aimed at improving
the prediction of HP.
A real-time implementation of the HyMeX-dedicated version of the Météo-France
AROME NWP system covering the whole WMed ran from 01 September 2012 to 15
March 2013 (Fourrié et al., 2015). The same system was used to perform an extensive
reanalysis of SOP1 exploiting observations from research instruments deployed during
the campaign in addition to the operational observations assimilated in real-time (Fourrié
et al., 2019). For that, processing of observations and systematic comparisons between
observations and AROME short-range forecasts were carried out for: i) ground-based
Lidar water vapour observations in Candillargues (BASIL) and at Menorca (WALI), ii)
airborne Lidar LEANDRE II water vapour observations along the SAFIRE/ATR42 flight
tracks, iii) high-resolution radiosoundings from operational sites in France and Spain and
HyMeX -dedicated radiosoundings launched during SOP1 over France and Italy, iv)
dropsondes observations and in-situ observations from the three research aircrafts, v)
reprocessed wind profiler observations, vi) reprocessed delays from more than 1000
GPS receivers over France, Spain, Portugal and Italy, vi) radar data from five AEMET



operational radars over Spain, vii) additional SST observations from ship and Argo
floats. The skill scores showed a better performance for the forecasts starting from the
re-analysis than those starting from the real-time AROME-WMED analysis. Data denial
experiments, for which one of the above-listed datasets was removed from the reanalysis
at a time, clearly showed the benefit of assimilating the reprocessed GPS ground-based
zenithal total delays as shown in Figure 9.
This result was confirmed in other studies. Lindskog et al. (2017) demonstrated the
benefits of GPS assimilation to the forecast quality. Bastin et al. (2019) pointed out that
the general overestimation of low values of IWV in RCM models over Europe was
reduced when using a nudging technique to assimilate IWV information. Caldas-Alvarez
and Khodayar (2020) highlighted the positive impact exerted by moisture corrections on
precipitating convection and the chain of processes leading to it across scales.
Furthermore, the implementation of nudging methodologies to exploit non-conventional
observations, such as rainfall estimates from remote sensing, provided positive results
in applications to both nowcasting and short-term meteo-hydrological forecasting
(Davolio et al., 2017; Poletti et al., 2019).
The potential of several new types of observations within cloudy and precipitating
systems have been also investigated. As a first step towards assimilation, "observation
operators", which consist in simulating observations from model outputs, have been
developed. In the framework of HyMeX, a dual-polarization weather radar simulator has
been developed in the post-processing part of the Meso-NH mesoscale model (Augros
et al., 2015). An observation operator for the airborne Rasta reflectivity observations has
also been developed (Borderies et al., 2019a). The impact of the assimilation of RASTA
data on AROME-WMED analyses and forecasts has been assessed. IOP7a results
indicated an improvement in the predicted wind at short-term ranges (2 and 3 hours) and
in the 12-hour precipitation forecasts. Over a longer cycled period, a slightly positive
improvement in the 6-, 9- and 12-hour precipitation forecasts of heavy rainfall has been
demonstrated (Borderies et al., 2019a). The assimilation of RASTA reflectivity data in
AROME-WMED over the whole SOP1 period resulted in an improvement of rainfall
forecasts even larger when wind was jointly assimilated (Borderies et al., 2019b).
Finally, HyMeX has fostered the inception of a collaboration between CNRM (France)
and CNR-ISAC (Italy) concerning the assimilation of radar data. The assimilation of radar
reflectivity factor together with lightning, showed a significant and positive impact on the
short-term precipitation forecasts (Federico et al., 2019).






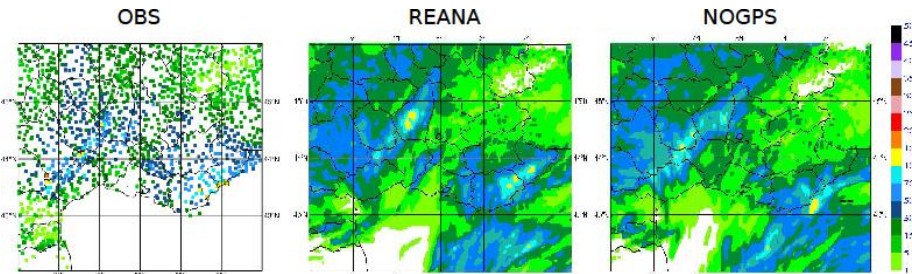

**Figure 9:** 24 hour accumulated precipitation (mm/24hr) between 06:00 UTC, 26 Oct. and 06:00 UTC, 27 Oct. 2012 over southern France (zoom over Cévennes area); observations (left panel), REANA (middle panel) and NOGPS (right panel) simulations.

### 4.4 Predictability and ensemble forecasts

Despite advances in numerical modelling and data assimilation, the prediction of HP and related floods remains challenging because predictability of intense convective systems is limited, and user expectations are very high given the impact of HPEs. Ensemble prediction techniques can provide solutions through the elaboration of probabilistic HPE warnings. Until the 2010s, regional ensemble prediction systems were mainly limited by the large computational costs of increasing the members resolution and the ensemble size. Ensemble forecasts become even more useful when post-processing techniques are applied to the precipitation fields, using statistical methods such as regression or analogues (Diomede et al., 2014), with some difficulties due to the geographically complex forecast error structures of Mediterranean precipitation, and to the need of preparing very long reforecast datasets in order to adequately sample the statistical behaviour of HPEs.

With the availability of more powerful computational resources, operational regional models started to reach the kilometric resolution leading to physically more realistic convection-permitting ensemble prediction systems (CPEPS). Studies of HPE events (Nuissier et al., 2016) have shown the added value of CPEPS over deterministic approaches or lower resolution ensembles. The enhanced exchange of validation datasets during HyMeX facilitated the objective verification of this kind of result in several ensemble studies such as Roux et al. (2019).

Several studies in the framework of HyMeX have demonstrated that, besides sensitivity to synoptic scale forcing represented by lateral boundary condition (LBC) perturbations, CPEPS systems were sensitive to multiple error sources, which had to be sampled as





specific perturbation in the parameterization schemes. Some sensitivity of ensemble spread to the model physics (turbulence and microphysics schemes) was demonstrated in Hally et al. (2014), who used the HyMeX IOP6 and IOP7a forecast and observation dataset to show that LBC perturbations cannot be neglected, and that the relative importance of LBC and physics uncertainties is case-dependent, with physics uncertainty being more significant during weakly forced convective events. Vié et al. (2012) explored the impact of microphysical processes on CPEPS spread and found a relationship between precipitation evaporation and the uncertainty of cold pool formation, which can be relevant to predict the correct location of HPEs. Bouttier et al. (2015) found a beneficial impact of randomly perturbing surface fields such as SST or soil moisture: the high density of HyMeX SOP1 data gave statistical significance to these results, because the objective verification of ensembles at high resolution requires large observational datasets.

Besides developing physics perturbation techniques, other ensemble approaches were tested in HyMeX case studies, based on different models or parameterization schemes. This multiphysics or multi-model technique was shown to be relevant to HPE events in several studies, such as Davolio et al. (2013), Hally et al. (2015), Ravazzani et al. (2016). Compared to other sources of uncertainty, the maximum impact of physics, multiphysics or surface perturbations tends to be observed at forecast ranges between a few hours and about one day, after which the CPEPS behaviour is usually dominated by the LBCs.

The specification of the ensemble LBCs can be optimized in terms of the HPE forecasts: Nuissier et al. (2012) showed that LBC member selection from a global ensemble, using a clustering technique, improves over a random selection. Marsigli et al. (2014) demonstrated that direct nesting into the ECMWF ensemble, instead of using an intermediate model, is beneficial despite the large resolution jump between the global and CP ensemble.

Initial condition perturbations using ensemble data assimilation systems were studied in Vié et al. (2012) and Bouttier et al. (2016). They found that initial condition perturbations are critical to achieve a correct CPEPS ensemble spread, typically during the first twelve hours of prediction, after which other perturbation sources (LBCs, surface and stochastic model perturbations) tend to dominate.

Verification of ensemble forecasts of HPEs can be overtaken using probabilistic scores, It greatly benefits from the large amount of observations available during the SOPs. Ensemble predictions can also be evaluated by their ability to drive ensembles of hydrological runoff models. Indeed, it was confirmed during HyMeX that although hydrological models suffer for their own uncertainties (Edouard et al., 2018), precipitation forecast



errors are the main sources of uncertainty for flood prediction. Many studies dealing with
CPEPS also exploited precipitation forecasts to drive flood prediction systems. Among
others, Roux et al. (2019) pointed out that enhancing the spread of HPE precipitation
forecasts tends to help flood warnings by improving the detection of extreme HPE sce-
narios.
An emerging application of CPEPS forecasts consists in investigating the physical mech-
anisms that drive HPE events (and, possibly, the reasons behind forecast failures), as
exemplified in the USA by Nielsen and Schumacher (2016). In the HyMeX framework,
beside the above-mentioned studies on physical perturbations, importance of orograph-
ically driven low-level flows was confirmed using CPEPS in Hally et al. (2014) and Nuis-
sier et al. (2016). Figure 10 shows an example for the IOP16 case.


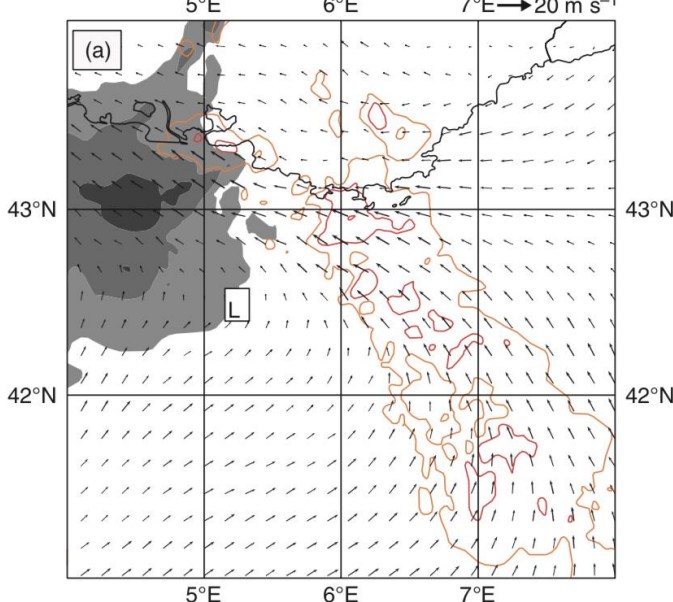



**Figure 10:** Lowest quartile of the mean sea-level pressure (shading), mean 10m wind
(arrows) and mean moisture flux at 925hPa (solid lines: 80 and 100 g m s⁻²) valid at 12:00
UTC, 26 Oct. 2012 for the AROME-EPS ensemble. (credits: Nuissier et al., 2016)


**5. Discussion**



The spatial complexity of the Mediterranean region, the intricacy of the dynamical and physical processes involved including the multiple interactions across scales, as well as the technical and observational limitations in the past have made HP understanding and modelling in the Mediterranean region a very challenging issue. To try to advance in this direction the Hydrological Cycle in the Mediterranean Experiment (HyMeX, 2010-2020) has put a major effort in investigating the predictability and evolution of extreme weather events. Within this framework and profiting from the state-of-the-art observational datasets and modelling capabilities lately available and developed within the programme, important achievements towards improved understanding of the mechanisms leading to HP in the WMed have been accomplished. In this paper we review the main advances and lessons learned during HyMeX, including results emerged from cross-disciplinary studies.

The unprecedented richness of observations and numerical experiments during HyMeX led to major achievements and the identification of primary needs for an improved understanding and predictability of HPE. Our better comprehension of the moist convergence role on MCS initiation over the sea, the first-time airborne observations of WCB, the high space-time resolution measurements of the 3-D fields of water vapour, or the testing of new convection-permitting ensembles, which provided new insights on HPE predictability and of forecast error sources, among others illustrate the main accomplishments achieved during HyMeX. Parallel to this, observational, modelling and knowledge gaps have been identified clearly indicating the needs for future applications. Sensitivity to soils and sea surface conditions with impacts on high-resolution forecasts pointed out the need to build and/or improve high-resolution coupled systems able to represent the full evolution of the soil and ocean conditions. The need of a higher number of observations, for example collected over the sea, thermodynamic profile measurements and wind on a high space-time resolution would have a major impact on forecasting capabilities, through the initialization of modelling systems, data assimilation and the definition of improved parameterization schemes for turbulence and convection. Also, open questions remain regarding access to large samples of HPE reforecasts, the representation of model error processes specific to HPE systems and persisting shortcomings in the real-time prediction of extreme precipitation events for flood warnings, among others. Furthermore, coordinated research efforts will be needed to address topics of multi-scale interactions, from large-scale dynamics to microphysical processes.





Along with the aforementioned achievements and demands, the continuous
collaboration between scientific communities, e.g., oceanographers and meteorologists,
and among scientific teams (ST) has been a priority and a success of the project. In fact
coordinated efforts, in particular with the ST-lightning (Lightning and atmospheric
electricity), ST-TIP (Towards integrated prediction of heavy precipitation, flash-floods
and impacts), ST-ffv (Flash-floods and social vulnerabilities), and ST-medcyclones
(Mediterranean Cyclones), through the development and use of common observation
and modelling tools, and by sharing results and expertise, helped each other towards
common goals. As illustration, aiming at a better understanding of processes leading to
flash floods, as well as at their accurate modelling and forecasting, the ST-ffv actively
contributed to the improvement of heavy rainfall prediction. Several recent
multidisciplinary studies investigated the possibility to have an integrated modelling
approach from heavy rainfall forecasting, to discharge prediction, to social impact.
Methodologies of postflood field surveys based on interdisciplinary collaborations
between hydrologists and social scientists have been proposed (Ruin et al., 2014; Borga
et al., 2019). For instance, Papagiannaki et al. (2017) investigated the link between HP
and impacts on the flash flood that occurred in October 2015 in Attica. The survey
responses provided insights into risk perception and behavioral reactions relative to the
space-time distribution of rainfall. Different possibilities of improving hydrometeorological
forecasts have also been tested (Roux et al., 2020), pointing out the added value of
ensemble strategies with respect to deterministic forecasts. Large meteorological
ensemble spreads also allowed better threshold exceedance detection for flood warning.
Furthermore, the rapid increase of total lightning flash rates has been found to be an
important predictor for severe weather phenomena (e.g., Wu et al., 2018), which is
closely related to the rapid increase of graupel concentration and updraft volumes in the
mixed-phase layers of deep convective systems. Furthermore, many studies in the
framework of the ST-lightning have been devoted to the examination of the relationship
of lightning activity with microphysical properties of convective systems along their life
cycle. During HyMeX SOP1, the HyMeX lightning mapping array network (HyLMA; Defer
et al., 2015) was operated to locate and characterize the 3D lightning activity over the
Cévenne-Vivarais area at flash, storm, and regional scales. This unique and
comprehensive lightning data clearly showed the large potential for improving our
knowledge about the cloud microphysics, especially the distribution and evolution of ice
hydrometeors by taking advantage of cloud electrification. This challenging subject is
expected to be further addressed in near future.





The increased computational capacity, the development of high-resolution convection-permitting models and the availability of state-of-the-art observations have demonstrated to be of pivotal importance to attain a better understanding and modelling of HP in the last decade. Nevertheless, still the availability of observational data on the analysis of, e.g., small-scale processes, remains a limiting factor that challenges progress in process understanding and model evaluation, particularly when trying to underpin results from high-resolution model experiments with corresponding observations. Additionally, evaluation is expected to continue in those recent investigation lines developed within the HyMeX programme, which have already demonstrated their usefulness for advancing prediction or knowledge of HPE, such as CPEPS systems or the development and availability of fully coupled soil-vegetation-atmosphere-ocean models. Furthermore, the benefit of working under the umbrella of a long-lasting international experiment such as HyMeX allowed an effective and fruitful exchange of information on challenges, experiences, and goals, exploited through numerous multidisciplinary research activities. These interdisciplinary efforts were crucial to come towards improved understanding of the mechanisms leading to HP in the WMed. The links and networks originated in the framework of HyMeX must continue and even be enlarged in the future to progress together towards more integrated approaches. Novel integrated multidisciplinary research partnerships based on cross-sectional collaborations will be indeed needed to bridge more efficient research on impacts.

**Code and Data Availability**

Given this is a review publication, the data and code availability are provided in each of the referenced publications.

**Author contributions**

All authors collaborated and contributed to drafting, reviewing, and editing the paper. In particular, SK coordinated the effort and wrote the original draft; SD contributed to the reviewing of the low-level dynamical processes; PDG contributed to the reviewing of the observational capabilities; CLB contributed to the reviewing of the air-sea coupling;  EF contributed to the reviewing of the large-scale dynamics; NF contributed to the reviewing of data assimilation; KOL contributed to the reviewing of the low-level dynamical processes; DR contributed to the reviewing of improved parameterizations; BV



contributed to the reviewing of the microphysics; FB contributed to the reviewing of the
predictability and ensemble forecast.

**Competing Interests**


The authors declare that they have no conflict of interest.

**Acknowledgements**


This work is a contribution to the HyMeX programme, supported in France by MISTRALS
(Météo-France, CNRS, INRAE) and the Agence Nationale de la Recherche (ANR
MUSIC grant ANR-14-CE01-0014, ANR IODA-MED grant ANR-11-BS56-0005). We
would like to thank all HyMeX contributors, and particularly to all members of the
scientific-team heavy precipitation (ST-HP), more than 100 in the last 10 years, which
have effectively participated in advancing knowledge regarding heavy precipitation in the
Mediterranean region. Without their work this study would not be possible.  We further
acknowledge the HyMeX database developers and all data providers.
We also thank all HyMeX scientific team coordinators for the close cooperation during
these years. We acknowledge Hélène Roux and Eric Defer for their input on research
activities of the ST-ffv and ST-lightning. The authors thank Marie-Noëlle Bouin (CNRM
& LOPS) who provided us MESO-NH simulation data related to IOP16a sensitivity to
waves. The contribution of Paolo Di Girolamo to this work was possible based on the
support from the Italian Ministry for Education, University and Research under the Grant
OT4CLIMA and FISR2019_01711 CONCERNING.

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
