# Peer review of "Overview towards improved understanding of the mechanisms leading to heavy precipitation in the Western Mediterranean: lessons learned from HyMeX"

_Atmospheric Chemistry and Physics, 2021_

## Author Comment (AC1)

**Answers to Reviewer1**

*"Overview towards improved understanding of the mechanisms leading to heavy precipitation in the Western Mediterranean: lessons learned from HyMeX"* by Samira Khodayar et al., Atmos. Chem. Phys. Discuss., https://doi.org/10.5194/acp-2021-350-RC1, 2021

Dear reviewer1

Thank you for your effort in reviewing the present manuscript. We appreciate all your comments and suggestions. We have considered all of them. Please find below a detailed answer to all your suggestions.

With kind regards

Samira Khodayar on behalf of all co-authors
* * *
This paper presents an overview of the main mechanisms associated with heavy precipitation over the Mediterranean. In particular, it reviews the main outcomes from the HyMex international project and their field campaigns. After a general introduction and a description of the observational and modelling infrastructure, the paper encompasses many of the key mechanisms leading to heavy precipitation. In general, all mesoscale and microphysical aspects are very well covered with the correct highlight of the results coming from the HyMex observation campaign.

The only aspect which in my view could have been further expanded, given also the relevance in the genesis of heavy precipitation, is their relation with the large-scale dynamics (section 3.1). Amongst the important literature on this topic, like Massacand et al. 1998 and Martius et al. 2008, it is worth mentioning a few very recent papers like 1) De Vries 2020, 2) Mastrantonas et al. 2021, 3) Grazzini et al. Part II 2021, which are showing robust and quantitative relations and dynamical linkages with large-scale precursors. But I understand that the focus of the current review is more on the mesoscale and local interaction so take my suggestion as not compulsory but as a simple indication. You decide whether it is necessary to expand the section. Other than that I think the manuscript is ready for publication after very minor changes reported below.

References:

1) Vries, A. J. D., 2020: A global climatological perspective on the importance of Rossby wave breaking and intense moisture transport for extreme precipitation events. Weather and Climate Dynamics.

2) Mastrantonas, N, Herrera-Lormendez, P, Magnusson, L, Pappenberger, F, Matschullat, J. Extreme precipitation events in the Mediterranean: Spatiotemporal characteristics and connection to large-scale atmospheric flow patterns. Int J Climatol. 2021; 41: 2710– 2728. https://doi.org/10.1002/joc.6985

3) Grazzini, F, Fragkoulidis, G, Teubler, F, Wirth, V, Craig, GC. Extreme precipitation events over northern Italy. Part II: Dynamical precursors. Q J R Meteorol Soc. 2021; 147: 1237– 1257. https://doi.org/10.1002/qj.3969

**Thank you for this comment. We agree that much more can be said about the relation with the large-scale dynamics, however, as the reviewer correctly points out the focus of the current review is on the mesoscale. Moreover, the paper is already lengthy. Nevertheless, to clarify this relevant point to the readers we have included a paragraph in the introduction, additionally referencing the papers suggested.**

**In the new document L112-121** *"Nevertheless, in the last few years, relevant knowledge has been gained in the field. Vries (2020) presented for the first time a global and systematic climatological analysis of the Rossby Wave Breaking (RWB) and intense moisture transport, and their linkage to extreme precipitation events (EPEs) in several regions, with the findings of this study contributing to an improved understanding of the atmospheric processes that lead to EPEs. Mastrantonas et al (2021) demonstrated that a clustering combination of sea level pressure (SLP) and geopotential height at 500 hPa (Z500), increases by more than three the conditional probability of EPEs, which could result critical for extended-range forecasts. Grazzini et al. (2021) further investigated the relation between EPEs and Rossby Wave Packets (RWPs), showing the evolution and properties of precursor RWPs key for the categorization of EPEs."*

Minor changes

- On page 9, line 291, you mention "comma-shaped" cloud coverage. You did not explain the meaning but hinting some implication of that. The explanation comes later, on pages 10 lines 324-327. It would be preferable to introduce first the relevant concepts.

**Thank you for this comment. We rephrased lines 290-293 of the original submission as follows:**

**"During the night from the 25th to the 26th of October, and in the following day, several MCSs formed under the influence of the cyclone and within its "comma-shaped" cloud coverage. Such cloud coverage is typically found in mid-latitude storms and owes its shape to warm conveyor belts (WCB; Eckhardt et al. 2004; Madonna et al. 2014), i.e., the airstreams that ascend slantwise over the cyclone warm front. All MCSs showed a quasi-stationary behavior, forming first over the sea, between the eastern Spanish coast and the Balearic Islands (Duffourg et al., 2016), and afterwards over the Gulf of Lion where they induced large amounts of precipitation over sea during morning hours."**

**Accordingly, we changed lines 322-328 as follows:**

**"Therefore, most intense Mediterranean cyclones are baroclinic systems with frontal structures and associated airstreams such as dry air intrusions and WCB (Ziv et al. 2009; Flaounas et al. 2015a). In particular WCBs are associated with stratiform, but also with convective rainfall due to embedded convection..."**

**Eckhardt, S., Stohl, A., Wernli, H., James, P., Forster, C., and Spichtinger, N.: A 15-Year Climatology of Warm Conveyor Belts, J. Climate, 17, 218–237, 2004**

**Madonna, E., Wernli, H., Joos, H., and Martius, O.: Warm conveyor belts in the ERA-Interim**

**dataset (1979–2010). Part I: Climatology and potential vorticity evolution, J. Climate, 27, 3–26, doi:10.1175/jcli-d-12-00720.1, 2014.**

- For the benefit of readers non used to European geography, it would be helpful to show CI, NEI, CO regions (only defined in the text) delimited on a geographical map

**Thank you for this comment, we have included following your suggestion boxes with the location of these subdomains in Figure 1 of the maunscript**

---

## Author Comment (AC2)

**Answers to Reviewer2**

*"Overview towards improved understanding of the mechanisms leading to heavy precipitation in the Western Mediterranean: lessons learned from HyMeX" by Samira Khodayar et al., Atmos. Chem. Phys. Discuss., https://doi.org/10.5194/acp-2021-350-RC1, 2021*

Dear reviewer2

Thank you for your comments and suggestions to this manuscript. We appreciate your effort. We have considered all your suggestions and implemented them in the manuscript. Please find below a detailed answer to all your suggestions.

With kind regards

Samira Khodayar on behalf of all co-authors
* * *
The paper "Overview towards improved understanding of the mechanisms leading to heavy precipitation in the Western Mediterranean: lessons learned from HyMeX", by Samira Khodayar et al. is a good review about the present understanding of the heavy precipitation in the Western Mediterranean, mostly oriented to work recently done in connection with the field phase "SOP1" (2012) of the "Hydrological cycle in the Mediterranean Experiment" (HyMeX), but also including references to research about the matter, not strictly based in HyMex; even significant work done before the HyMeX initiation is referenced here. By this way a more complete vision about the problem is achieved. This review is not the first about HyMeX-SOP1, but it is timely now, because 2020 was indicated as the end of HyMeX. Ducrocq et al (2014), can be seen as an initial review, and a special issue (2016) of the Quarterly Journal of the Royal Meteorological Society, introduced by Ducrocq et al. (2016; https://doi.org/10.1002/qj.2856), can be considered as a second one. In any case, since 2016 to present days' significant research results have been added. A positive aspect of this review is the team of authors that has been formed: their different specialisations cover many aspects of the problem and this permit a wider and also more precise vision of it. The number of commented references is really large. The list of references occupies more than 24 pages of a total of 63 pages. This means a hard and valuable work, but the list of references is not complete. In fact, it would be almost impossible to construct a complete list. In summary, I consider this paper is good enough, it is useful and it is an interesting tool to face the problem of the West-Mediterranean heavy rain in many aspects, and therefore I recommend its publication, almost as it is. In the following there are a few comments that can be taken into account by the authors, although the consideration of all of them is not strictly mandatory to publish the paper

Details

–Line 154 mentions 16 IOPs during the campaign SOP1: the total of IOPs was 20 (see Ducrocq et al, 2014)

**Thank you for this correction.**

-The measurement site named BA (Balearic Islands) not only included Menorca (were specific facilities were installed), but also Mallorca, with operational radar and radiosounding stations (lines 180-181)

**Corrected. Thank you for the information.**

-In figure 1, I don't understand what the colour of each radar station means

**All radars are shown in green. The other colors indicate the different instrumentation. All radars from the Hymex database have been included in this plot.**

-By line 320, it seems that the authors consider that only breaking Rossby waves (cut-off lows) can induce Mediterranean cyclogeneses, but also open troughs can do it

**Thank you for this comment. The "upper tropospheric filaments of air masses of high potential vorticity (PV)" are long known to be direct results of Rossby wave breaking. Depending on the vertical level, these PV filaments may be seen as cut-off lows over the Mediterranean or trough-like systems that extend beyond the limits of the region. In these regards, we agree with the Reviewer that we should be more precise. We rephrased lines 319-321 as follows:**

**"Mediterranean cyclogenesis is typically preceded by the intrusion of upper tropospheric systems such as troughs and cut-off lows. Such systems are typically shown to be direct results of Rossby wave breaking over the Atlantic ocean and be related with high potential voticity values that trigger cyclogenesis in the Mediterranean due to baroclinic instability (Grams et al., 2011; Raveh-Rubin and Flaounas, 2017)"**

-Fig. 2a, b. The meaning of the colours is not clear to me

**This figure follows the methods and uses the dataset of Flaounas et al. 2016. We revised the caption to be clearer.**

**Figure 2: (a) Sea level pressure (black contours every 3 hPa, outer contour is set at 1005 hPa). Coloured lines show the pressure level of the WCB air masses, related to the cyclone. WCBs are calculated using the ECMWF analyses and correspond to air mass trajctories that present an ascent of more than 500 hPa within 48 hours. In panel (a), we only show the 48-hour the trajectories that correspond to WCBs and where air masses are located close to the cyclones centre at 12:00 UTC, 26 Oct. 2012, between 680 and 720 hPa. i.e. within the ascending part of the WCBs (line segments of cyan colours). (b) as in (a), but for 12:00 UTC, 27 Oct. 2012. (c) Colours show daily accumulation of precipitation on 26 Oct. 2012 taken from 3B42 of TRMM. (d) as in (c) for 27 Oct. Datasets and methods for all panels are detailed in Flaounas et al. (2016).**

-In section 3.2.1, another interesting reference for water origin of heavy rain in Western Mediterranean is Insua-Costa et al (2019; https://doi.org/10.5194/hess-23-3885-2019)

**Thank you, we included this reference.**

**L416-420 of the new document: "Recent advances in this topic showed that the evaporation in the western Mediterranean, in the central Mediterranean, in the North Atlantic, and the advection from the tropical and subtropical Atlantic and Africa constitute the four moisture sources which could explain most of the accumulated precipitation in the WMed (Insua-Costa et al. 2019)."**

-With regard to section 3.3.1, IOP8, the most important event that affected Spain during SOP1, and also IOP15 and others, are analysed in Jansà et al (2014; https://doi.org/10.3369/tethys.2014.11.03). [Note also that Ferreti et al (2014; https://doi.org/10.5194/hess-18-1953-2014) is a parallel paper on the cases that affected Italy]

**Thank you for this comment. Given the relevance and the thematic of the articles, reviews of SOP1 over Spain and Italy, respectively, we have included them in the introduction of the manuscript**

**"…This issue is one of the main objectives of the HyMeX international programme, and of its associated first special observation period (SOP1; Ducrocq et al., 2014; Jansa et al., 2014; Ferreti et al., 2014), …"**

-Regarding section 3.3.2, the convergence associated to a cold pool boundary was already highlighted as a continued triggering convection mechanism in Western-Mediterranean heavy rain in Ramis et al (1994; https://doi.org/10.1002/met.5060010404)

**Included in L629-630 of the new document: "Ramis et al. (1994) already pointed out the convergence associated to a cold pool boundary as a continued triggering convection mechanism in the WMed."**

-Fig. 6 is not clear enough to me

**We have improved the description of the figure in the text and the correspondent legend to make it clearer to the reader.**

**" … This is shown in Figure 6, which presents the distribution of the simulated and observed RASTA reflectivities, sorted by altitude. Data from IOPs 6 and 16 were combined and classified in bins of altitude and reflectivity, and the number of events in each category was normalized by the total number of data points to provide the colored frequency."**

**Figure 6: Distribution of simulated (left: ICE3, right : LIMA) and observed (center) Comparison of observed and simulated RASTA reflectivities sorted by temperature, , merging data from IOPs 6 and 16a (From Taufour et al. 20186).**

-With regard to section 4.3, the Arome-WMed reanalyses, made with assimilation of all available added observations, permitted to detect a secondary cyclone that produced severe weather in Menorca during IOP18. See Carrió et al (2020; https://doi.org/10.1016/j.atmosres.2020.104983)

**Thank you, we included this reference.**

**L945-947 of the new document "Additional benefits were identified such as the detection of a secondary cyclone producing severe weather in Menorca during IOP 18 (Carrió et al. 2020)."**

-Continuing with 4.3, Campins et al (2016; https://doi.org/10.1002/qj.2737) studied the impact in the forecasting of the assimilation of some extra observations during SOP1

**Thank you, we included this reference**

**L925-928 of the new document: "Campins and Navascués (2016) evaluated the impact of targeted observations on HIRLAM forecasts during HyMeX-SOP1 showing that the assimilation of radiosoundings and Advanced TIROS Operational Vertical Sounder (ATOVS) satellite observations clearly improve the first-guess quality over land and sea sensitive areas respectively. "**